# Molecularly engineered supramolecular fluorescent chemodosimeter for measuring epinephrine dynamics

Yudan Zhao, Yuxiao Mei ✉, Zhichao Liu, Jing Sun ® & Yang Tian ® ✉

Accurately visualizing epinephrine (EP) activity is essential for understanding its physiological functions and pathological processes in brain. However, to the best of our knowledge, reliable, rapid, and specific measurement of EP dynamics at cellular and in vivo level hasn't been previously reported. Herein, we report the probe for EP imaging and biosensing in neurons and living brain of freely behaving animals, based on creating a series of supramolecular fluorescent chemodosimeters through host-guest interaction. The optimized chemodosimeter enables real-time imaging and quantifying of EP with high specificity, sensitivity, signal-to-noise ratio, and rapid kinetics (~240 ms) in neurons, brain tissues and zebrafish. More significantly, we demonstrate real-time monitoring of EP in 26 regions within deep brain of freely behaving male mice, unraveling an augmented EP concentration in the amygdala, thalamus, hypothalamus, hippocampus and striatum under fear-induced stress. These findings highlight our chemodosimeter as a powerful tool for precise measurements of EP dynamics in diverse model organisms.

Epinephrine (adrenaline or adrenalin, EP), plays an essential role in various physiological and pathological processes within the nervous system[1-3]. The dysfunction of adrenergic transmission is closely related to numerous neurodegenerative and psychiatric disorders, such as epilepsy, Alzheimer's disease (AD), and Parkinson's disease (PD)[4-10]. It is thus essential to specifically recognize and accurately quantify EP with high sensitivity, as well as to track EP dynamics across different brain regions. However, EP's structural similarity to other neurotransmitters, especially dopamine and norepinephrine, coupled with its rapid release kinetics at the rapid level, pose formidable challenges in developing probes that can specifically recognize and rapidly respond toward EP within the brain.

The ability to precise and rapid measure EP dynamics in vivo is of substantial interest for understanding its functions. Although several methods have been utilized to determine EP, including mass spectrometry[11,12], liquid chromatography[13], and capillary electrophoresis[14], these are not suitable for in vivo applications. Particularly, real-time monitoring and in situ tracking of EP dynamics within living cells and brains is still elusive. Therefore, there is an urgent need to develop powerful probes to achieve high specificity, high sensitivity, high spatiotemporal resolution for direct visualization of transient EP dynamics in neurons, and monitor EP dynamics in multiple brain regions in freely moving animals. Such progress hold promise for advancing the comprehension of the physiological and pathological functions of EP.

Herein, we reported a sensor based on designing and creating the supramolecular fluorescent probe for real-time imaging and quantifying of EP dynamics on a ~240 ms time scale in brain regions of mice (Fig. 1). Fluorescence microscopy-based real-time imaging has emerged as an ideal technology to detect various neurotransmitters, however the fluorescent probe for EP detection in living cells is unlikely to exist[15-20]. In this work, the supramolecular fluorescent chemodosimeters were designed and molecularly engineered by host-guest interaction. Host molecules, such as cucurbit[n]urils with different cavity sizes were coupled to styryl pyridinium-derived dye guest molecules containing boric acid moieties and reactive difluorophenol esters, while introducing methylene blue (MB) as a guest reference molecule to achieve quantitative detection[21-25]. The optimized

Shanghai Key Laboratory of Green Chemistry and Chemical Processes, School of Chemistry and Molecular Engineering, East China Normal University, Dongchuan Road 500, Shanghai, PR China. ✉e-mail: yxmei@chem.ecnu.edu.cn; ytian@chem.ecnu.edu.cn

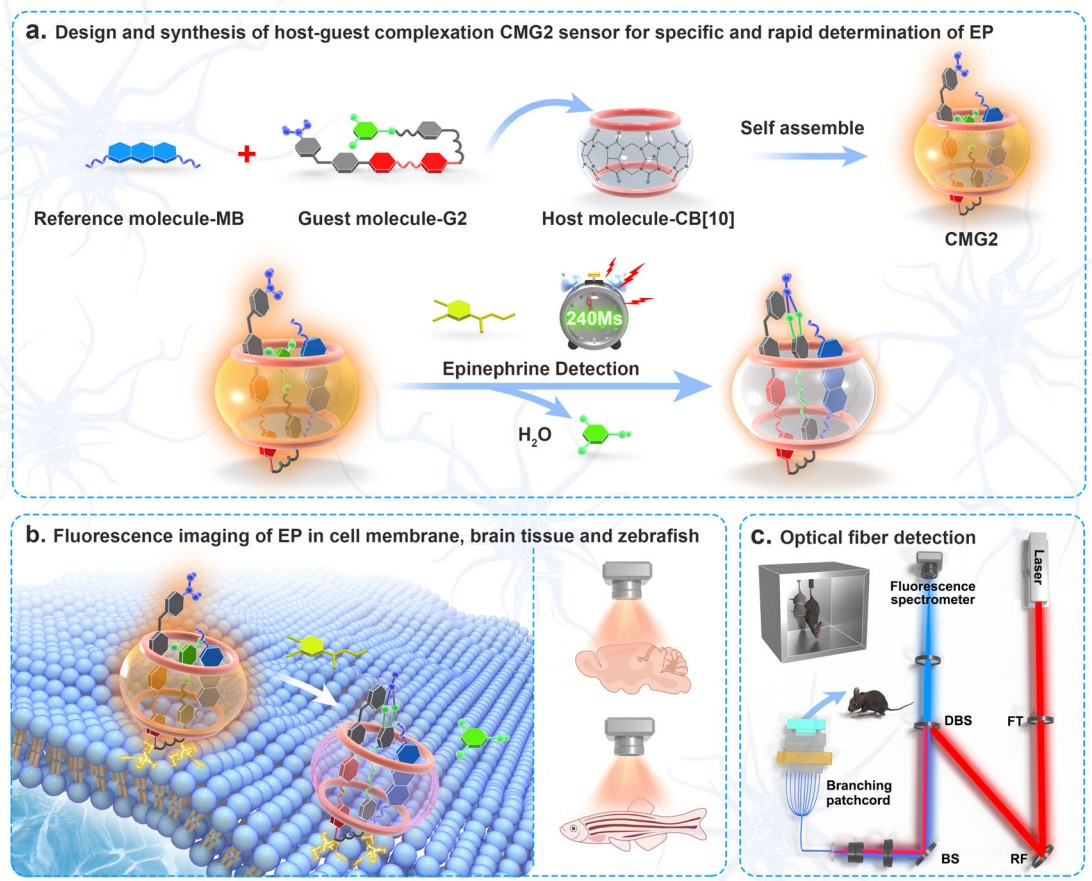

**Fig. 1 | The host-guest GMG2 chemodosimeter for visualizing and quantifying of EP in vitro and in vivo. a** The working principle of the developed and optimized CMG2 sensor for specific and rapid determination of EP. **b** Fluorescence imaging and real-time quantification of EP in cell membrane, brain tissue, and zebrafish. **c** Schematic diagram of fiber optic construction of multiple brain regions in mice.

chemodosimeter (named as CMG2) exhibited rapid response kinetics and high selectivity for EP against other neurotransmitters and amino acids. It was found that the concentration of dynamically released EP on the neuron membrane increased approximately ~35-fold within ~4.0 s after electrical stimulation. Fluorescence imaging of brain tissue slices showed that EP concentration significantly increased after electrical stimulation in the CA1, Cpu, and LD regions by ~1.96-fold, ~1.79-fold, and ~1.93-fold, respectively, highlighting heterogeneity in EP distribution across different brain regions. Additionally, fluorescence imaging of live juvenile zebrafish revealed a significant increase in EP concentration after electrical stimulation. Finally, real-time monitoring and quantifying of EP in 26 brain regions by a high-density fluorescent fiber array based on our developed supramolecular fluorescent chemodosimeter revealed that EP levels in the amygdala, thalamus, hypothalamus, hippocampus and striatum of mice predominantly increased under fear-induced stress.

## Results

### Design and synthesis of host–guest supramolecular fluorescent chemodosimeters for selective recognition toward EP with high temporal resolution

Ratiometric supramolecular fluorescent chemodosimeters were created for EP recognition with rapid response using host-guest synergy effects. As shown in Fig. 2a, three guest molecules of styryl pyridinium dyes (G1, G2 and G3) were synthesized. These molecules, G1–G3, featured a boronic acid group designed for interaction with catechol-type neurotransmitters through the condensation reaction, and an active

fluorinated phenyl ester that can react with monoamine-type neurotransmitters through an ester-amide exchange reaction[26–28]. The combination of piperazine with styrylpyridinium salt fluorophores could produce an intramolecular charge transfer effect (ICT), resulting in enhancement of fluorescence intensity and quantum yield[29–31]. In addition, a linker with different alkyl chain length was incorporated to adjust the conformation[32]. Host molecules CB[n]s with diverse cavity sizes (CB[6], CB[7], CB[8] and CB[10]) were introduced to enhance the confinement effect (Fig. 2a), while MB served as the guest fluorescence reference molecule for quantitative detection[33]. The pyridinium group in G1–G3 and the nitrogen heterocyclic group in MB enable the self-assembly with the cavities of CB[n]s, forming host-guest systems[34]. All the guest molecules were characterized by nuclear magnetic resonance (NMR), mass spectrometry (MS) and high-performance liquid chromatography (HPLC) (Details are available in Supplementary Information, Supplementary Figs. 1–65 and Supplementary Table 1).

The host-guest complex self-assembly generates the supramolecular fluorescent chemodosimeters (Fig. 2b). Initially, the influence of host molecules on the response performance was investigated by fluorescence titration using G2 as a representative with sufficient molecular flexibility to form a folded conformation. Four host molecules varied in size, and the details including portal diameter, cavity diameter, cavity volume, outer diameter and height were displayed in Supplementary Table 2[35,36]. The larger cavity sizes of CBs enable them to form 1:1 binary complexes (CB[7]) and even 1:1:1 heteroternary complexes (CB[8], CB[10]) with aromatic compounds[37,38]. When host molecules (CB[6], CB[7], CB[8] and CB[10]) were added to a

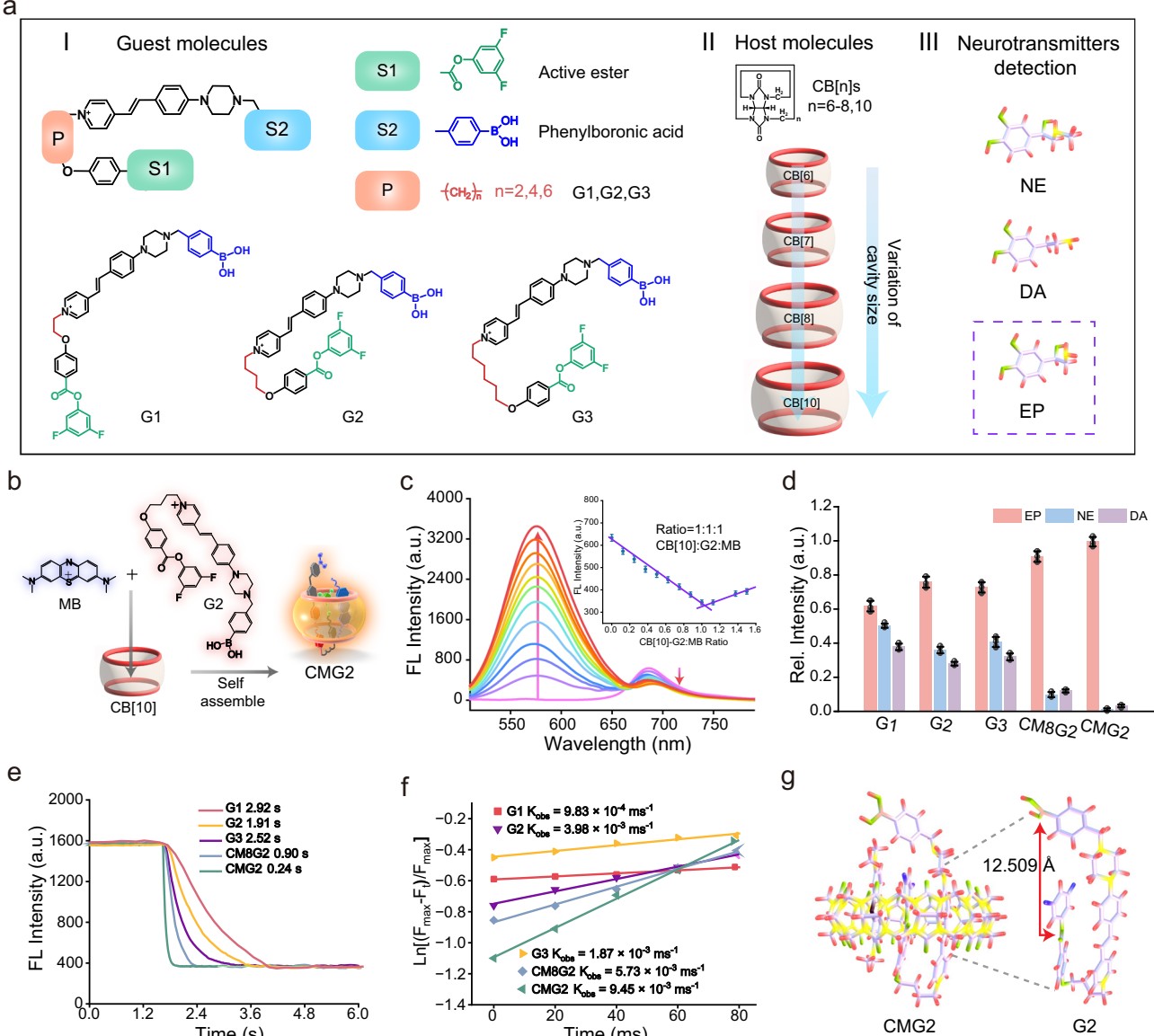

**Fig. 2 | Design and synthesis of host–guest supramolecular fluorescent chemodosimeters for selective and rapid response toward EP. a** Schematics of guest recognition molecules G1–G3 with varies alkyl chain lengths, host molecules CB[n]s with different cavity sizes, and neurotransmitters to be tested. **b** The self-assembly of the developed and optimized CMG2 supramolecular fluorescent sensor for EP. **c** Fluorescence titration spectra of MB (10 μM) toward addition of CB[10]-G2 (concentration ratio, 1:1) with different concentration (0–15 μM) in PBS buffer (10 mM, pH = 7.4) containing 0.05% DMSO; Illustration: CB[10]-G2 with MB assembled molar ratio. Data are presented as mean ± S.D. Error bars: S.D., n = 5 independent experiments. **d** Interference of G1, G2, G3, CM8G2 and CMG2 on NE and DA (Set the response value of CMG2 for EP to 1.0). Data are presented as

mean ± S.D. Error bars: S.D., n = 5 independent experiments. **e** Kinetic fluorescence responses of the G1–G3 (each 10 μM) and the assembled host-guest supramolecular fluorescent sensor toward addition of 5 μM EP in PBS (10 mM, pH 7.4). **f** Reaction rate constants ($k_{obs}$) were measured for the G1–G3 and the assembled host-guest supramolecular fluorescent sensor in response to EP. The pseudo-first-order rate constants ($k_{obs}$) obtained from the slopes of plot of $\ln[(F_{max}-F_t)/F_{max}]$ vs time. $F_{max}$ is the maximum fluorescence intensity during the measurement time, $F_t$ is the fluorescence intensity at the corresponding time points ($F_{max}$, $F_t$: The fluorescent emission wavelength is 570 nm). **g** The molecular structure of CMG2 after DFT optimization and the quantification of the distance between two recognition sites of G2. The above-mentioned source data are provided as a Source Data file.

mixture of guest molecules G2 and MB (concentration ratio = 1:1), G2 and MB showed the fluorescent emission at 570 nm ($F_{570}$) and 695 nm ($F_{695}$), respectively (Supplementary Fig. 66). Both the emission at $F_{570}$ and $F_{695}$ changed upon the addition of $CB^8$ or $CB^{10}$, while no simultaneous change was observed after adding $CB^6$ or $CB^7$, indicating that $CB^8$ and $CB^{10}$ could encapsulate the two guest molecules to form supramolecular fluorescent chemodosimeters, CM8G2 and CMG2. The corresponding Job's plot clearly suggested a 1:2 $CB^8(CB^{10})$-MB, 1:2 $CB^8$-G2, 1:1 $CB^{10}$-G2 and 1:1:1 $CB^8$ ($CB^{10}$)-G2-MB binding stoichiometry (Fig. 2c and Supplementary Fig. 67), which is in good agreement with previous reports[39,40].

We then conducted selectivity test with three guest molecules. G1, G2 and G3 showed no interference from various amino acids, but were significantly interfered by norepinephrine (NE) and dopamine (DA) (Supplementary Fig. 68). G1 was interfered by up to 81% and 61% for NE and DA, respectively, G2 was interfered by 47% and 37% for NE and DA, respectively, and G3 was interfered by 56% and 44% for NE and DA, respectively (Fig. 2d). Subsequently, the selectivity of the corresponding chemodosimeters toward EP were examined. When compared to G2 alone, the introduction of host molecule significantly improved its selectivity for EP, among which CMG2 demonstrated the highest selectivity for specific recognition of EP (Supplementary

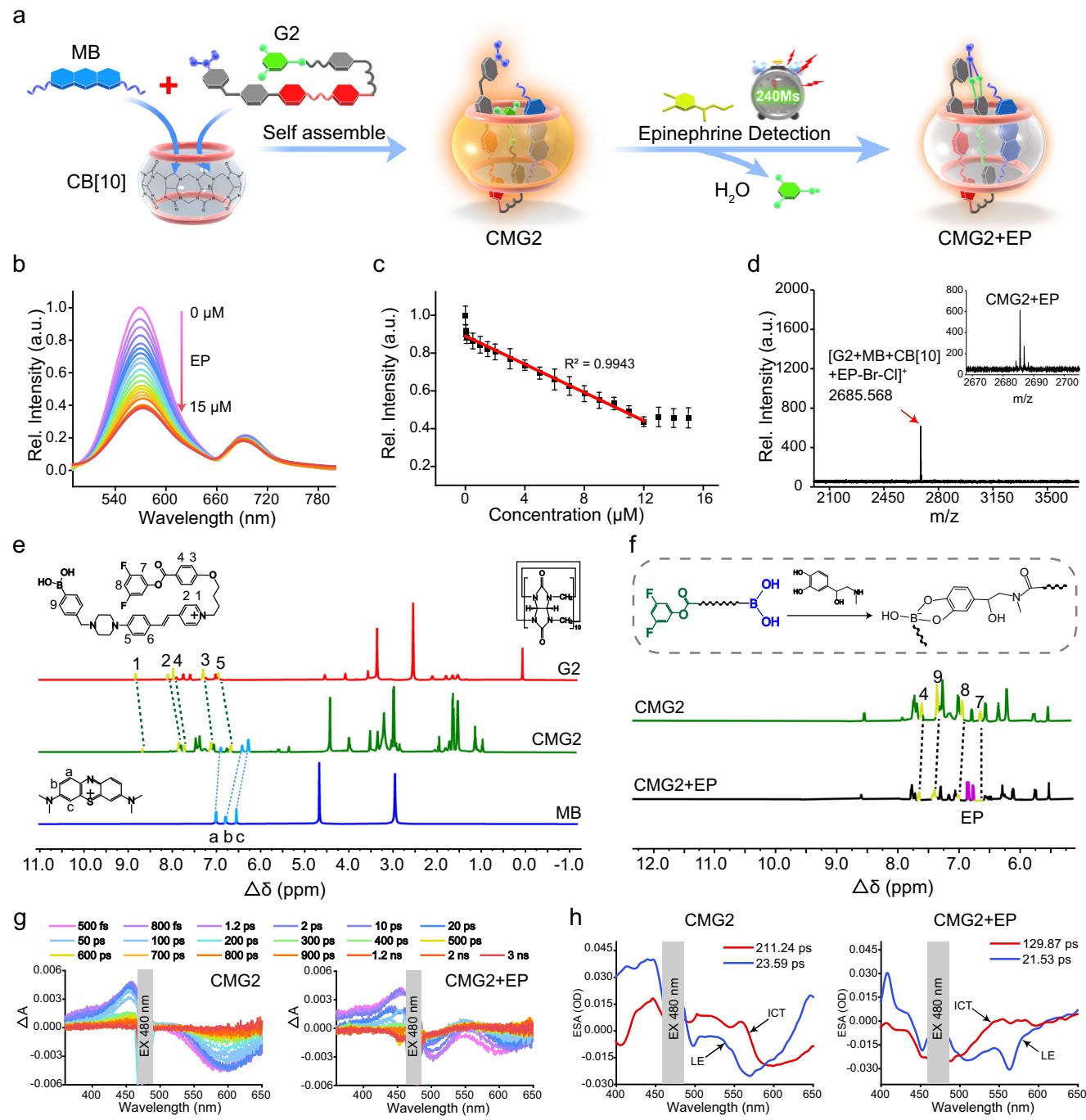

**Fig. 3 | Fluorescence titration of CMG2 toward EP and mechanism evaluation.**
**a** The working principle of the developed and optimized CMG2 for specific and rapid determination of EP. **b** Fluorescence spectra of 10.0 μM CMG2 with the addition of EP at different concentrations (0– 15 μM) in cell lysis buffer (10 mM, pH = 7.4) containing 0.05% DMSO excited at 480 nm. **c** Relative fluorescence intensity of the CMG2 versus EP concentration (0–15 μM). Data are presented as mean ± S.D. Error bars: S.D., $n$ = 5 independent experiments. **d** Maldi-TOF mass spectrometry of CMG2 + EP. **e** $^1$HNMR spectra of G2, MB and CMG2 ([G2] = [MB] = [CB$^{10}$] = [CMG2] = 5 mM). **f** $^1$HNMR spectra of CMG2 and CMG2 assembled with EP ([CMG2] = [EP] = 5 mM, in DMSO-$d_6$: D$_2$O = 1: 1 at 298 K). **g** Transient absorption spectroscopy for CMG2 (left), and CMG2 + EP (right). **h** Principal spectral components from the global analysis of time-resolved absorption spectra from 480 nm excitation for CMG2 (left), and CMG2 + EP (right). The above-mentioned source data are provided as a Source Data file.

Fig. 68). CMG2 exhibited almost no responses for other neurotransmitters. Especially, the interference of CMG2 on NE and DA was significantly reduced to 1.1% and 3.2%. Furthermore, no obvious fluorescence changes were obtained in competition tests when determining EP upon the addition of potential interferences (other amino acids, carbohydrates, metal ions and reactive oxygen species) (Supplementary Fig. 69). The fluorescence response kinetics of the sensing reaction for the probes was measured by using a rapid-mixing

stopped-flow technique with a finite mixing time of less than 8 ms (Supplementary Fig. 70). It was found that the fluorescence intensity at $F_{570}$ decreased immediately after addition of EP and reached a plateau at different times with a sequence of G2 (1.91 s) <G3 (2.52 s) <G1 (2.92 s) (Fig. 2e). The reaction rate constant ($k_{obs}$) values for G1–G3 in response to EP were $9.83 \times 10^{-4}$ ms$^{-1}$, $3.98 \times 10^{-3}$ ms$^{-1}$ and $1.87 \times 10^{-3}$ ms$^{-1}$ (Fig. 2f), indicating a quick response between EP and the guest molecules. Therefore, G2 was screened as the optimized guest molecules for the

construction of supramolecular fluorescent chemodosimeters. When host molecules, CB[10] or CB[8] were introduced to the system, similar behavior for the response kinetics of the developed supramolecular fluorescent chemodosimeters were observed with a sequence of CMG2 (240 ms) <CM8G2 (900 ms) <<G2 (1.91 s). The $k_{obs}$ for CMG2 and CM8G2 was $9.45 \times 10^{-3}\,ms^{-1}$ and $5.73 \times 10^{-3}\,ms^{-1}$, respectively, indicating faster response to EP after complexing with the host molecules.

To further elucidate the improved performance, the first-principle calculation based on the Density Functional Theory (DFT) was conducted. Of note, the pKa of EP, DA, and NE are 45.5749, 42.4725, and 41.9999, respectively (Supplementary Tables 3–5). Electrostatic Potential Energy (ESP) distribution and Natural Population Analysis (NPA) charge calculation showed that the electronegativity of the nitrogen atoms in EP, DA and NE is −0.653 (EP) < −0.641 (DA) < −0.636 (NE) (Supplementary Fig. 71). These findings suggest that EP is more reactive to the active fluorinated phenyl ester site of CMG2 than NE and DA. The structures of free G1–G3 and supramolecular ternary complex CMG2 were optimized by DFT calculation (Fig. 2g and Supplementary Fig. 72). G1 was in a relatively extended conformation due to the larger molecular bond tension and steric hindrance, while G2 and G3 had sufficient molecular flexibility to form a folded conformation. Moreover, when G2 was assembled with the host molecules, the distance between the recognition sites could be further shortened due to the confinement of the cavity. The distance between the recognition sites in free G1–G3 and complex CMG2 was further quantified by theoretical calculation (Supplementary Table 6). With the introduction of CB[10], the distance was shortened from 14.204 Å to 12.509 Å. Since CMG2 probe has the fastest response rate to EP and the optimum specificity for EP recognition, CMG2 was selected as the optimized supramolecular probe for subsequent imaging and sensing of living systems.

## Fluorescence titration of CMG2 toward EP and mechanism evaluation

The working principle of CMG2 for the determination of EP was shown in Fig. 3a. The fluorescence titration experiment was carried out in buffer solution for CMG2 upon addition of EP. As shown in Fig. 3b, $F_{570}$ channel (collected from 505 to 645 nm) decreased sharply with increasing concentration of EP, which almost reached the minimum when 12.0 μM of EP was added, while $F_{695}$ channel (collected from 665 to 750 nm) remained unchanged (five replicate experiments in Supplementary Fig. 73). The relative fluorescence intensity of CMG2 showed a good linear correlation ($R^2 = 0.9943$) with the concentration of EP varying from 0.02 to 12.0 μM. The regression equation was estimated as $F_{570/695} = 3.965 - 0.166$ [EP] μM (Supplementary Fig. 74). The limit of detection (LOD) was estimated to be $5.1 \pm 0.3$ nM ($n = 5$, S.D.) (Fig. 3c). According to Eqs. (1) and (2), the binding affinity between CMG2 and EP was calculated. The estimated binding affinity was $4.66 \times 10^4\,M^{-1}$[41,42] (Supplementary Fig. 75).

$$[HG] = 0.5 \left\{ \left( [H]_0 + [G]_0 + \frac{1}{Ka} \right) - \sqrt{\left( [H]_0 + [G]_0 + \frac{1}{Ka} \right)^2 - 4[H]_0[G]_0} \right\}$$

(1)

$$\frac{F}{F_0} = 1 - k[HG]/H_0$$

(2)

In which, $[H]_0$ is the total concentration of CMG2 used for the titration, $[G]_0$ is the total concentration of EP used for the titration, $K_a$ is the association constant. CMG2 showed long-term photostability in the absence (<1.3%) or presence (<1.9%) of EP at 480 nm irradiation for 8 h (Supplementary Fig. 76a). No significant variation of $F_{570}/F_{695}$ in the

absence (<1.5%) or presence (<2.1%) of EP was detected in the pH range of 6.0-10.0 (Supplementary Fig. 76b). All these results demonstrate that long-term stability of CMG2 for application in living systems.

To gain more insight into the self-assembly process, Matrix-Assisted Laser Desorption Ionization Time of Flight Mass Spectrometry (MALDI-TOF-MS) and NMR spectroscopy were used to investigate the host-guest binding and the corresponding stoichiometry. A new peak (mass-to-charge ratio, m/z = 1324.896) belonging to [CMG2-Br⁻-Cl⁻]²⁺ was found after mixing CB[10], MB and G2 (Supplementary Fig. 77), indicating the successful host-guest encapsulation. When adding EP, a new m/z peak of 2685.568, belonging to [CMG2 + EP-Br⁻-Cl⁻]⁺ was detected (Fig. 3d). To get detailed structural characterization of the supramolecular complex CMG2, the NMR resonances of G2 was assigned with the aid of 2D ¹H-¹H COSY NMR techniques (Supplementary Fig. 78). Based on these unequivocal assignments of the protons of G2, we were able to study the self-assembly behavior through NMR experiments. Compared with the spectrum of free MB and G2, the aromatic proton signals $H_{a,b,c}$ of MB were shifted upfield in the presence of CB[10] ($\Delta\delta = 0.15, 0.25, 0.18$ ppm for $H_a$, $H_b$, $H_c$, respectively). The proton signals $H_{1,2,3,4,5}$ of the pyridine and the aromatic region portion of G2 were shifted upfield ($\Delta\delta = 0.13, 0.17, 0.10, 0.11, 0.18$ ppm for $H_1$, $H_2$, $H_3$, $H_4$, $H_5$, respectively) (Fig. 3e). Such an upfield shifted behavior is due to strong shielding effect by CB[10], indicating the successful encapsulation by CB[10]. These findings were further confirmed by 2D ROESY NMR analysis with the NOE correlations between proton $H_x$ of CB[10] and proton $H_a$, $H_b$, $H_c$, $H_3$, $H_5$ of MB and G2 (Supplementary Fig. 79a). The couples of NOE signals between protons $H_1$-$H_3$ and $H_2$-$H_3$ of the G2 confirmed the steric proximity of the two phenyl groups, indicating a typical characteristic of folded conformation, which was consistent with the above results of response speed measurement and DFT calculation. Similarly, upon the addition of EP, secondary amines reacted with esters to form amide products, resulting in the weakening of $H_7$ and $H_8$ signals of G2 molecules. The electron cloud density of the benzene was weakened due to the lower electronegativity of the nitrogen atom than that of the oxygen atom, leading to downfield shift ($\Delta\delta = 0.08$ ppm) of the $H_6$ signal of G2. Meanwhile, the proton $H_9$ adjacent to the boronic acid group of G2 underwent a significant downfield shift with $\Delta\delta$ values of 0.07 ppm, indicating an increased shielding effect around these protons due to the electron-donating effect of the phenyl borate formed by the condensation reaction between boronic acid and catechol groups (Fig. 3f). 2D ROESY NMR spectroscopy further confirmed the correlations of $H_x$-$H_{II}$, $H_x$-$H_{III}$, $H_9$-$H_{III}$ ($H_{II}$, $H_{III}$: protons of EP) between CMG2 and EP (Supplementary Fig. 79b). Therefore, we concluded that CMG2 complexed with EP through two recognition sites of G2 and encapsulated EP in the cavity of CB[10].

To rationalize the changes in the luminescence of CMG2 after addition of EP, the fluorescence lifetime was initially investigated in PBS buffer solution under excitation at 480 nm. It was observed that the average fluorescence lifetime of CMG2 + EP (2.486 ns) increased upon the addition of EP when compared to that of pristine CMG2 (1.783 ns) (Supplementary Fig. 80). Femtosecond transient absorption (fsTA) measurements were also investigated, revealing two distinct excited states of CMG2: the locally excited state S1 (LE) and the intramolecular charge transfer state S1 (ICT) (Fig. 3g, h). After reaction with EP, the relaxation of the LE state to the ICT state was inhibited due to the formation of the borate ester structure. This change caused the ICT/LE relaxation ratio to shift from 0.78 to 0.17, leading to the quenching of the fluorescence of CMG2 at 570 nm.

## Fluorescence imaging and real-time quantification of EP in neurons, brain tissues, and zebrafish

To apply CMG2 for imaging and biosensing of EP in biological systems, the biocompatibility and cytotoxicity of CMG2 were evaluated by flow

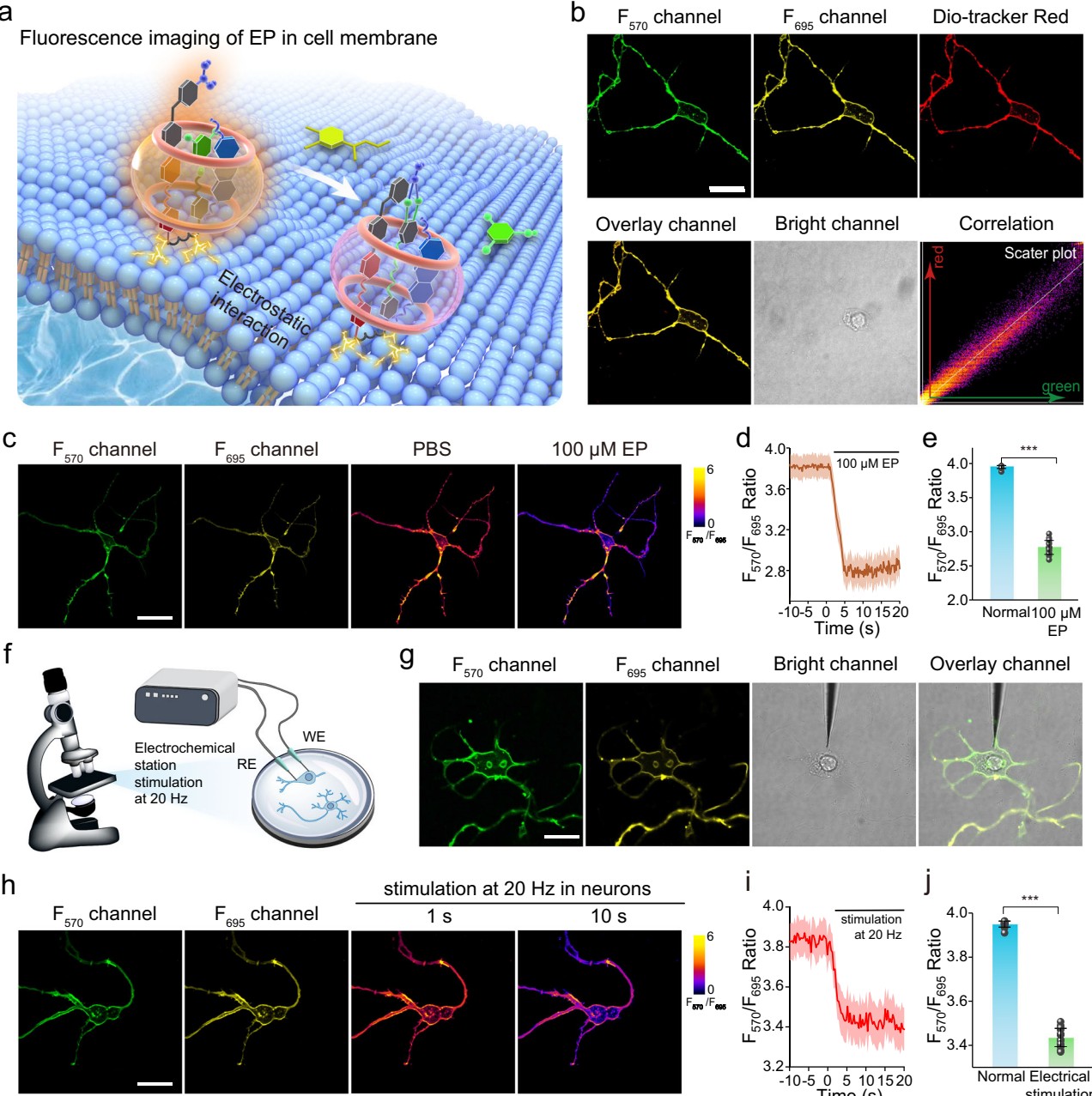

**Fig. 4 | Fluorescence imaging and real-time quantification of EP in neurons.**
**a** Fluorescence imaging and real-time quantification of EP in cell membrane.
**b** Confocal fluorescence images of neurons costained with CMG2 and a commercial membrane probe (Dio). Three independent experiments were repeated and similar results were obtained. Scale bar: 15 μm. **c** Representative images showing the fluorescence images of CMG2 at cell membrane and their response to 100 μm EP. Scale bar: 15 μm. **d** Representative traces of CMG2 in response to 100 μm EP. **e** Dynamic response summary of CMG2 in response to 100 μm EP ($n$ = 15 cells).
**f** Schematic diagram of electrically stimulated neurons. **g** Confocal fluorescence images of electrically stimulated neurons. Scale bar: 10 μm. **h** Time-lapse confocal fluorescence images of CMG2-incubated neurons after electrical stimulation. Scale bar: 15 μm. **i** Representative traces of CMG2 after electrical stimulation ($n$ = 15 cells). **j** Dynamic response summary of CMG2 after electrical stimulation. The above-mentioned data are all presented as mean ± S.D. Error bars: S.D., gray dots represent individual data points. Statistical significance is calculated with a two-tailed unpaired t-test and $P$ values are indicated (***$p$ < 0.001). Source data are provided as a Source Data file.

cytometry (FACS) analysis and MTT assay[43,44]. The viability of living neurons remained at ~90.7% after incubation with increasing concentrations of CMG2 (0, 10, 30 and 50 μM) for 24 h, showing good biocompatibility (Supplementary Fig. 81a, b). The MTT assay also exhibited high viabilities of neurons (above 85%) when incubating with CMG2 at various concentrations (0, 10, 20, 30, and 50 μM) for 24 h, indicating low cytotoxicity (Supplementary Fig. 81c). These results suggest that CMG2 has good biocompatible and low cytotoxicity. In addition, CMG2 could maintain photostability for up to 24 h in vivo,

confirming its suitability for imaging applications (Supplementary Fig. 81d).

The EP response dynamics at cell membrane was examined by the fluorescence imaging (Fig. 4a). The co-staining neurons with CMG2 and a commercial membrane dye Dio for 30 min showed a strong overlay between the fluorescence signals of $F_{570}$ channel of CMG2 and Dio, with a high Pearson correlation coefficient of 0.96, indicating excellent neuronal cytomembrane targeting ability (Fig. 4b). We hypothesized that the cationic pyridinium have electrostatic affinity

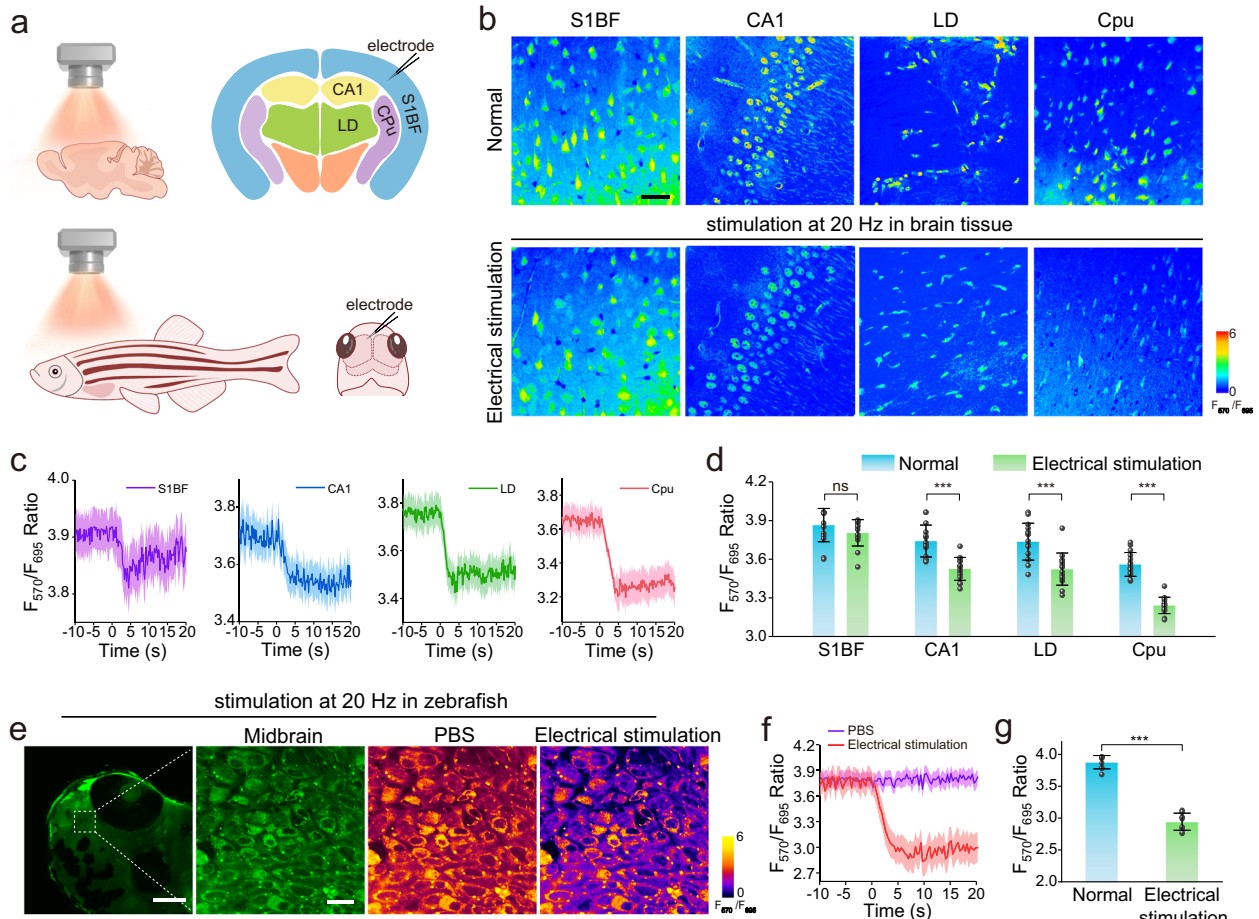

**Fig. 5 | Fluorescence imaging and real-time quantification of EP in brain tissues and zebrafish. a** Fluorescence imaging and real-time quantification of EP in brain tissue and zebrafish. **b** Fluorescence images of brain tissue slices from S1BF, CA1, LD and Cpu regions incubated with CMG2 after electrical stimulation. Scale bar: 75 μm. **c** Representative traces of CMG2 in S1BF, CA1, LD and Cpu after electrical stimulation. **d** Dynamic response summary of CMG2 after electrical stimulation in S1BF, CA1, LD and Cpu ($n = 15$ independent experiments). **e** Fluorescence images of zebrafish incubated with CMG2 after electrical stimulation. Scale bar: 130 μm

and 10 μm. **f** Representative traces of CMG2 in zebrafish after electrical stimulation. **g** Dynamic response summary of CMG2 after electrical stimulation in zebrafish ($n = 15$ independent experiments) The above-mentioned data are all presented as mean ± S.D. Error bars: S.D., gray dots represent individual data points. Statistical significance is calculated with a two-tailed unpaired t-test and $P$ values are indicated ($^{ns}p > 0.05$, $^*p < 0.05$, $^{**}p < 0.01$, and $^{***}p < 0.001$). Source data are provided as a Source Data file.

with the phosphate anion of the cell membrane surface, while the hydrophobic alkyl chain could embed into the cell membrane, contributing to the cell membrane targeting ability. The fluorescence signals of CMG2 remained stable (<8.5%) at cell membrane for 48 h, fulfilling the requirements for cell experiments. After 72 h incubation with cells, the targeting of CMG2 at cell membrane decreased, indicating that the probes were metabolized out of the cells (Supplementary Fig. 82).

The application of a saturating concentration of EP induced fluorescence decreases of CMG2 when compared to the PBS group, demonstrating the fluorescence response in neuronal cells (Fig. 4c–e and Supplementary Fig. 83). We further used CMG2 to investigate the EP response dynamics in neurons after external electrical stimulation (Fig. 4f, g). We first conducted control experiments to rule out the possibility that the fluorescence response was due to the electrochemical activity of the compounds. We applied a 20 Hz, 3 V sine wave (1.0 s) to the CMG2 solution and monitored the fluorescence intensity, which remained unchanged (Supplementary Figs. 84a, b). Additionally, NMR and MALDI-MS analyses of the solution showed no structural changes of CMG2 after external voltage (Supplementary Fig. 84c, d). After being treated with electrical stimulation (20 Hz, 3 V, sine wave for 1.0 s), the fluorescence intensity decreased significantly

within ~5.0 s, indicating a rapid increase of EP at the neuronal cytomembrane (Fig. 4h). Quantitatively analysis further confirmed this phenomenon (Fig. 4i, j), showing an increase in EP concentration from $0.09 \pm 0.02$ μM to $3.16 \pm 0.07$ μM within 4.0 s after electrical stimulation. However, the EP concentration in cell membrane remained almost constant after PBS buffer treatment (Supplementary Fig. 85). Therefore, our supramolecular fluorescent chemodosimeter successfully enables the monitoring of the EP response in neurons, allowing for the quantitative detection of EP concentration with precise cytomembrane-targeting and rapid spatiotemporal resolution.

Subsequently, we used CMG2 for imaging studies in brain tissue and zebrafish (Fig. 5a). We prepared acute brain tissue slices from four different regions of normal mice, including the cornu ammonis of primary somatosensory cortex (S1BF), hippocampus (CA1), caudate putamen (Cpu), and laterodorsal thalamic nucleus (LD). Real-time imaging and quantitative biosensing of EP were conducted in these brain regions, revealing the heterogeneity of EP. As shown in Fig. 5b, the concentrations of EP exhibited an increment upon electrical stimulation at 20 Hz. The average fluorescence ratios at $F_{570}/F_{695}$ channels significantly changed within specific time frames: from $3.74 \pm 0.12$ to $3.53 \pm 0.09$ in CA1 within 6.6 s, from $3.74 \pm 0.14$ to $3.52 \pm 0.12$ in LD

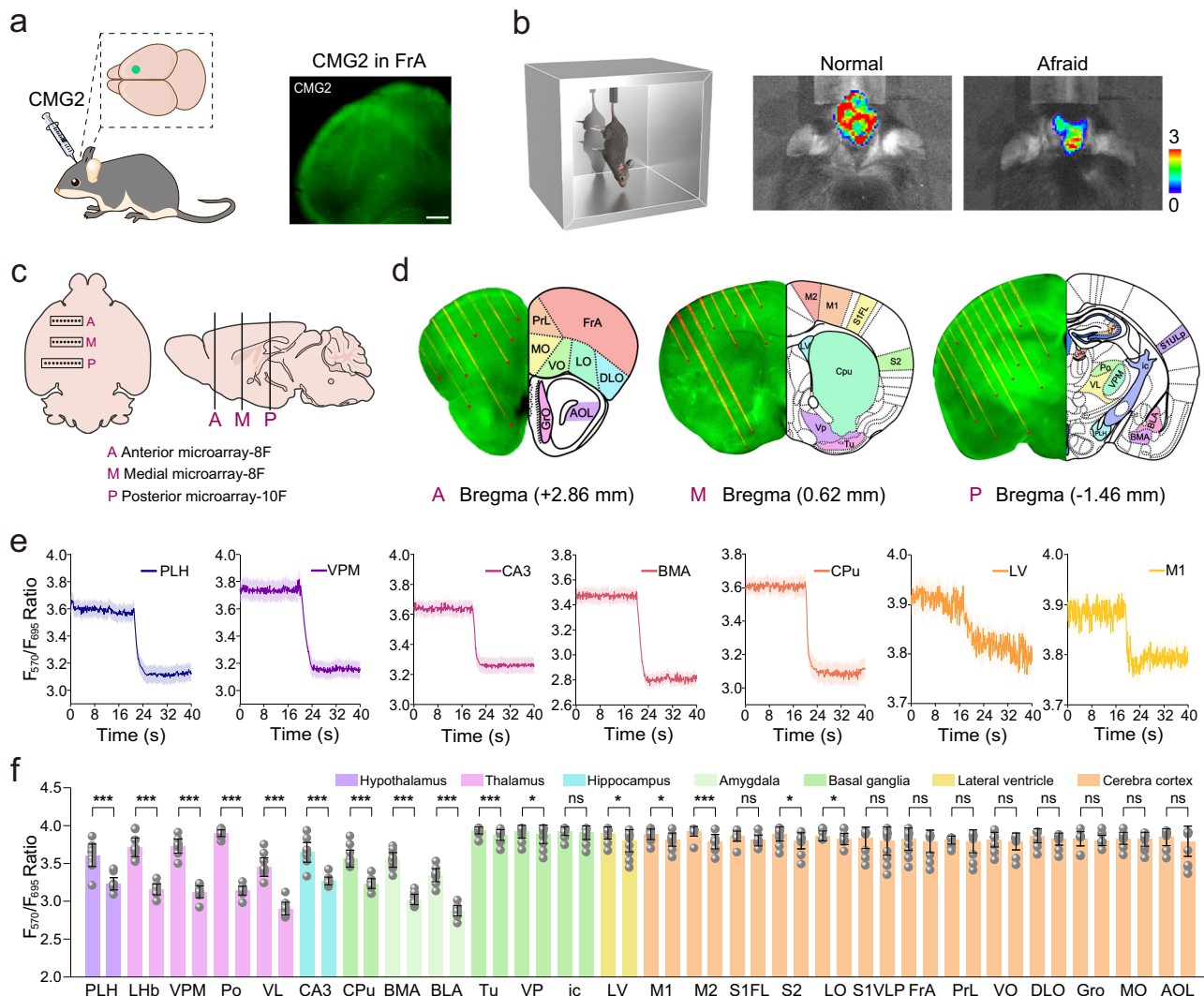

**Fig. 6 | Real-time monitoring and quantifying of EP in 26 brain regions.**
**a** Schematic illustration depicting probe injection and fluorescence imaging in vivo. Example image showing CMG2 in FrA in a coronal brain slice; Scale bar: 500 μm. **b** Schematic cartoon illustrating tail suspension experiments. In vivo imaging of normal and afraid-model mouse brains stained with CMG2. **c** An overhead and lateral view of the brain surface implanted with A multi-fiber microarray, including anterior (A) middle (M) and posterior (P). **d** Distribution of brain slices in different areas and three-dimensional remapping of Allen brain Atlas.

**e** Fluorescence signal F570/F695 ratio changes in seven representative brain regions with time after tail suspension stimulation. **f** Quantized maps of fluorescence signals in 26 different brain regions ($n = 15$ independent experiments). The above-mentioned data are all presented as mean ± S.D. Error bars: S.D., white dots represent individual data points. Statistical significance is calculated with a two-tailed unpaired t-test and P values are indicated ($^{ns}p > 0.05$, $*p < 0.05$, $**p < 0.01$, and $***p < 0.001$). Source data are provided as a Source Data file.

within 5.1 s, and from 3.56 ± 0.09 to 3.24 ± 0.06 in Cpu within 4.9 s. However, minimal changes were observed for EP in S1BF, with a ratio change from 3.87 ± 0.13 to 3.80 ± 0.10 (Fig. 5c, d). These results indicate that EP is more responsive to electrical stimulation in CA1, Cpu and LD compared to S1BF. We also performed 3D imaging of EP in living larval zebrafish (Supplementary Fig. 86), demonstrating that the value of $F_{570}/F_{695}$ changed from 3.88 ± 0.10 to 2.94 ± 0.13 within 7.8 s in response to electrical stimulation (Fig. 5e–g and Supplementary Fig. 87a). In contrast, the EP concentration in zebrafish remained relatively constant after PBS buffer treatment (Supplementary Fig. 87b). The results highlight the excellent imaging ability of the developed chemodosimeter, with high selectivity and rapid response speed towards EP.

## Real-time monitoring and quantifying of EP in 26 brain regions of mice

To ensure the suitability of CMG2 for in vivo imaging and biosensing of EP in different brain regions, the stability of CMG2 in brain, metabolic

pathways and biotoxicity were first examined. Fluorescence imaging showed that the injected CMG2 initially accumulated within the cells and then diffused outside the cells in the representative brain regions including cerebral cortex, hippocampus, thalamus and striatum (Supplementary Fig. 88a, b). Pharmacokinetic fluorescence imaging revealed that CMG2 was metabolized from the brain through the kidney and liver within 72 h after injection (Supplementary Fig. 88c). Moreover, hematoxylin-eosin (HE) staining results revealed negligible pathological changes in the brain and organ tissues even after 72 h of CMG2 injection (Supplementary Fig. 89), suggesting the safety and stability of CMG2.

The optical fiber implanted in brain was characterized, and the size of the optical fiber was measured (Supplementary Fig. 90a). The optical field distribution was estimated to be $5.12 \times 10^{-3}$ mm$^{-3}$ volume from the tapered fiber facet, consistent with the geometrical emission property of the experimental results (Supplementary Fig. 90b, c)[45,46]. After implanting the microarray into mouse brain (Supplementary Fig. 91a), 2,3,5-triphenyltetrazolium chloride (TTC) staining showed no

apparent trauma to the brain tissues after 3 h of microarray insertion (Supplementary Fig. 91b). Local field potential (LFP) signals proved that the developed microarray had minimal impact on neuron activities (Supplementary Fig. 91c). Confocal image of astrocytes stained for glial fibrillary acidic protein (GFAP) (green) surrounding an optical fiber shaft demonstrated no obvious tissue immune response (Supplementary Fig. 91d). These results indicate the good biocompatibility and biological safety of the developed microarrays and CMG2.

To detect EP levels in the brains of normal and tail suspension stimulated mice, CMG2 was administered in vivo (Fig. 6a, b). The co-staining brain tissue slices with CMG2 and a commercial membrane dye Dio for 30 min showed a strong overlay between the fluorescence signals of $F_{570}$ channel of CMG2 and Dio, with a high Pearson correlation coefficient of 0.95, indicating excellent cytomembrane targeting ability (Supplementary Fig. 92). A multi-channel fiber fluorescence spectroscopy was established to imaging and biosensing of EP in different brain regions. In three mice, a total of 26 fibers were implanted in different positions in one hemisphere, including anterior (A), middle (M) and posterior (P) region. The top and side views of the brain surface implanted with the multi-fiber microarray are shown in Fig. 6c. The selected brain regions belong to different neurobiological networks and consist of the following regions: Frontal association cortex (FrA), prelimbic cortex (PrL), Medial orbital cortex (MO), Orbitoventral cortex (VO), lateral orbital cortex (LO), Dorsolateral orbital cortex (DLO), Granulosa layer of olfactory bulb (GrO),Anterior lateral olfactory area (ALO), Primary somatosensory cortex superior labial area (S1VLp), primary motor cortex (M1), Secondary motor cortex (M2), Anterior limb area of primary somatosensory cortex (S1FL), Secondary somatosensory cortex (S2), field CA3 of the hippocampus (CA3), lateral ventricle (LV), internal capsule (ic), caudate putamen (Cpu), olfactory tubercle (Tu), ventral pallidum (VP), basolateral amygdaloid nucleus (BLA), basomedial amygdaloid nucleus (BMA), lateral hypothalamus (PLH), posterior nucleus of thalamus (PO), ventrolateral thalamic nucleus (VL), ventral postromedial nucleus (VPM), lateral habenular nucleus (LHb). Accurate localization of the developed fiber arrays into the target brain regions was confirmed by 3D reconstruction, which depicted the 3D location of the fiber tips and the neuronal labeling in front of these fiber tips in the brain (Fig. 6d)[47]. These brain regions encompass seven distinct structures, including the cerebral cortex, hippocampus, ventricles, thalamus, striatum, hypothalamus, and amygdala.

EP is a neurotransmitter associated with stress response and is known to be involved in emotional reactions of tension and fear[48]. To investigate EP dynamics in different brain regions in response to tail suspension stimulation, we employed the designed multi-channel fiber fluorescence spectroscopy system to examine the levels of EP in 26 specific brain areas of mice after injection of CMG2 into these regions while the mice were subjected to tail hanging. It was found that the levels of EP in amygdala (BMA, BLA), thalamus (LHb, VPM, PO and VL), hypothalamus (PLH), hippocampus (CA3) and striatum (Cpu) were obviously increased in response to tail suspension stimulation (Fig. 6e, f). The level of EP in the cerebral cortex (M2) was also slightly elevated, while relatively negligible changes in EP concentration were obtained in the other 16 brain areas (<2.8%). The increased electrophysiological signals in mice treated by tail suspension stimulation imply enhanced neuronal activity. These findings indicate that EP concentrations in the amygdala, thalamus, hypothalamus and hippocampus of mice were predominantly elevated under fear-induced stress, highlighting the role of these regions in stress response.

## Discussion

In summary, we report the molecularly engineered supramolecular fluorescent chemodosimeter for real-time imaging and quantitatively analysis of EP dynamics based on host-guest interactions. A family of

chemodosimeters were synthesized and characterized. The optimized CMG2 can be used both in vitro and in vivo to monitor EP activity with high affinity, high sensitivity, and fast kinetics (~240 ms). We were able to demonstrate that CMG2 detects EP dynamics after electrical stimulation in neurons, brain slices and zebrafish. Notably, we successfully monitored the fast EP response and quantify the concentration with precise cytomembrane-targeting and milliseconds-level spatiotemporal resolution. Furthermore, real-time monitoring of EP in 26 different regions in deep brain of freely behaving mice was achieved by home-made multi-channel fiber fluorescence spectroscopy system. We observed an augmented EP concentration in the amygdala, thalamus, hypothalamus, hippocampus and striatum in mice under fear-induced stress compared to normal mice. Therefore, our investigation has opened up a new way to real-time monitoring EP concentrations, distributions, and dynamic changes from neuron to whole brain, which helps us to understand the physiological and pathological processes of EP-related brain events. The present work further offers a methodology to design and synthesize the chemodosimeters for specific and sensitive determination of neurotransmitters, amino acids, and proteins in neurons, tissues, and brains of mice.

## Methods

### Materials

All chemicals and reagents were purchased from Sinopharm Chemical Reagent Co. Ltd. (Shanghai, China) and were used without further purification unless stated specifically. Neurobasal medium, trypsin and B27 were purchased from Thermo Fisher Scientific (U.S.A.).

### Syntheses

The synthetic procedures and details of the compounds mentioned in this report can be found in the Supplementary Information (Scheme S1).

### Instruments and kinetics study procedures

Nuclear magnetic resonance (NMR) spectra, including $^1$H NMR and $^{13}$C NMR, were acquired using a Bruker 500 MHz spectrometer, while 2D ROESY NMR spectra were recorded on a Bruker 600 MHz spectrometer (Bruker, Germany). High-resolution mass spectrometry (HRMS) analysis was conducted on a Bruker ESI time-of-flight mass spectrometer (Germany). Absorption and fluorescence spectra were obtained using a rectangular quartz cell (10 × 10 × 45 mm) on a Hitachi UH-5300 spectrometer (Japan) and a Hitachi F-4600 fluorescence spectrometer (Japan), respectively. To investigate the rapid fluorescence response of the probes to EP, a stopped-flow accessory with a pneumatic drive system (SFA-20, HI-Tech, TgK Scientific, United Kingdom) was employed. Fluorescence confocal imaging was performed using a Leica TCS-SP8 fluorescence microscope, equipped with an oil immersion objective (40×). Apoptosis assays were conducted using a FACSCalibur flow cytometer (Becton, Dickinson and Company, USA). All experiments were carried out at 298 K, unless otherwise specified.

### Culture of mouse cortical neurons

All mice involved in the experiment were male. All animal care and in vivo procedures were conducted in accordance with the guidelines set by the Animal Care and Use Committee of East China Normal University (approval no. m20240425, Shanghai, China). Mice were housed under standard conditions with free access to water and food, maintained on a 12-hour light-dark cycle. The Animal Care and Use Committee of East China Normal University, Shanghai, China supplied C57BL/6 wild-type (WT) mice. Newborn C57BL/6 wild-type mice, aged within 24 h, were anesthetized with halothane, and their brains were swiftly extracted and immersed in Hanks' balanced salt solution (HBSS, free of $Mg^{2+}$ and $Ca^{2+}$) on ice. Cortical tissue was swiftly dissociated and incubated in papain at 37 °C for 15 min, after which it was mechanically dispersed into poly-D-lysine-coated 35 mm Petri dishes at a density of

$1 \times 10^6$ cells per dish. Neurons were cultured in neurobasal medium supplemented with L-glutamine and B27 (37 °C, 5% $CO_2$, 95% $O_2$), with medium changes occurring three times per week. Wild-type (WT) zebrafish were also utilized in this study.

## Apoptosis assay and cytotoxicity analysis

For the apoptosis assay, pre-incubated neurons were exposed to varying concentrations of CMG2 (0, 10, 30, and 50 μM) for 24 h. Following removal of the culture medium, cells were harvested using EDTA-free trypsin. After washing three times with PBS (1 mL each), the cells were re-suspended in 300 μL of binding buffer and incubated with FITC-Annexin V and propidium iodide (PI) to label apoptotic and necrotic cells, respectively. The flow cytometry data were processed using FlowJo software (version X 10.0.7 R2). For cytotoxicity analysis, neurons pre-incubated in 96-well plates were treated with different concentrations of CMG2 (0, 10, 20, 30, and 50 μM) for 24 h. 20 μL of MTT solution was added to each well. After a 4-hour incubation, the mixture was removed, and 80 μL of DMSO was added to dissolve the formazan crystals. Following a 5-minute shake, absorbance at 490 nm was measured, and cell viability was calculated using the formula: cell viability (%) = absorbance of the experimental group/absorbance of the blank control group × 100%.

## Preparation and imaging of mouse brain tissue slices

All mice used in this experiment were male. Fresh brain tissue sections were obtained from 5-month-old C57BL/6 mice, purchased from the Laboratory Animal Center of the Chinese Academy of Sciences. Fresh brain slices, approximately 400 μm thick, were prepared using a German Leica VT3000 vibrating blade microtome. The slicing process was conducted in ice-cold artificial cerebrospinal fluid (ACSF), containing NaCl (124.0 mM), KCl (3.0 mM), $NaHCO_3$ (26.0 mM), $NaH_2PO_4$ (1.24 mM), $MgSO_4$ (8.0 mM), $CaCl_2$ (0.1 mM), and D-glucose (10.0 mM), under a 95% $O_2$ and 5% $CO_2$ atmosphere. The tissue sections were then transferred to ACSF containing 20.0 μM CMG2 and incubated at 37 °C for 60 min, with ACSF maintained under a 95% $O_2$ and 5% $CO_2$ environment. After incubation, the sections were washed with ACSF at least three times before imaging. Finally, stained sections were observed using a TCS-SP8 confocal laser scanning microscope equipped with a laser.

## Statistics and reproducibility

The times of some experiments repeated independently with similar results were stated in the legends. For other experiments (such as micrographs), three independent experiments were conducted, yielding similar results. Data are expressed as mean values ± standard deviation (S.D.), calculated using Microsoft Excel 2016. Statistical analysis is performed with an unpaired two-tailed Student's $t$ test using IBM SPSS 27 statistical software, followed by post hoc tests for multiple comparisons. $P$ value < 0.05 is considered statistically significant ($^{ns}p > 0.05$, $^*p \leq 0.05$, $^{**}p \leq 0.01$, $^{***}p \leq 0.001$). Each legend is labeled with a $P$-value, and the exact $p$-value and confidence interval are provided in the source data file. The n values, as indicated within the Fig. legends, represent biologically independent samples or independent experiments. Derived statistics are based on the analysis of averaged values across biological replicates, rather than technical replicates.

## Theoretical calculation

The theoretical calculations were performed via the Gaussian 16 package. Geometry optimizations and frequency analysis were performed at the M062X-D3/def2-SVP level of theory with the SMD of water. The subsequent natural population analysis (NPA) for the optimized structures was calculated using the def2-TZVP basis set at the same functional level. The pKa values (in the solution phase) were calculated from the free energy change over a thermodynamic cycle method[49]. Specifically, the Gibbs Energy for the gas phase dissociation of HA$_{(g)}$ and A$^-_{(g)}$ was calculated by the CBS-QB3 compound energy method. The Gibbs Energy for the solvation of HA$_{(aq)}$ and A$^-_{(aq)}$ was calculated by the m062x/6-31 g(d) method. The Gibbs Energy for H$^+_{(g)}$ and H$^+_{(aq)}$ were −6.28 kcal/mol and −265.9 kcal/mol, respectively from the best estimate values[50–52]. The solution phase free energy change ($\Delta G_{aq}$) was finally calculated is calculated using the combination of the above free energies. For all atoms the 6–31 G(d) Pople basis set was used. All of the optimized geometries mentioned were built by Gaussview 6.0.

## Reporting summary

Further information on research design is available in the Nature Portfolio Reporting Summary linked to this article.

## Data availability

All data supporting the findings of this study are available in this paper, Supplementary Information and from corresponding authors upon request. Source data are provided with this paper.

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

## Acknowledgements

This work was supported by the National Key Research and Development Program of China (2022YFF0710000 to Y.T.), the National Natural Science Foundation of China (21811540027 and 22393930 to Y.T.), the Innovation Program of Shanghai Municipal Education Commission (201701070005E00020 to Y.T.), and 2022 Shanghai "Science and Technology Innovation Action Plan" Fundamental Research Project (22JC1401200 to Y.T.) and Fundamental Research Funds for the Central Universities (to Y.T.). The authors also thank the Materials Characterization Center of East China Normal University for help with cell imaging.

## Author contributions

Y.T. and Y.X.M. conceived and designed the experiments. Y.D.Z. performed the experiments and wrote the manuscript. Z.C.L. helped with the bioimaging experiments. J.S. modified the manuscript. All the authors discussed the results and commented on the manuscript.

## Competing interests

The authors declare no competing interests.
