## [Transparent Peer Review file · Nature Communications]

Molecularly engineered supramolecular fluorescent chemodosimeter for measuring epinephrine dynamics

Corresponding Author: Professor Yang Tian

Version 0:

Reviewer comments:

Reviewer #1

(Remarks to the Author)

The real-time dynamics of epinephrine (EP) is highly relevant to decipher complex brain activities. However, there are currently no probes, including fluorescent protein probes and small molecule probes, that enable rapid detection of EP. To overcome this dilemma, Tian and coworkers reported a first supramolecular fluorescent chemsensor (CMG2) for real-time imaging and quantitative biosensing of EP dynamics in freely behaving animals. The overall performance of CMG2 showed high affinity, sensitivity, and fast kinetics (~ 240 ms) for monitoring EP activity *in vitro* and *in vivo*. Importantly, such a chemsensor could achieve real-time monitoring EP concentrations, distributions, and dynamic changes from neuron to whole brain. These data and results are particularly intriguing and critically important in the field of brain science. Overall, this paper can be considered for the publication after minor revision.

- 1) What is the rationale design behind this structure, despite the boronic acid recognition site and active fluorinated phenyl ester? What's the function of piperazine in the guest molecules? Please provide a sentence for the rationale behind this choice.
- 2) The authors should point out the physiological concentrations of EP in the manuscript to demonstrate the applicability of the designed probes.
- 3) There are some errors in chemical structures for guest molecules in SI, e.g., the reagent of di-bromide used to synthesize intermediate 3 should contain 6 methylene groups not 4.
- 4) What is the scale bar of Figure S80c?

Reviewer #2

(Remarks to the Author)

In this manuscript, the authors report self-assembled supramolecular host-guest complexes for the fluorescence-based detection of epinephrine (aka adrenaline). The complexes can be excited around 475 nm and its emission can be detected in the range from ca. 540 nm to ca. 720 nm rendering it suitable for bioimaging. The latter is demonstrated with neuronal cells, zebrafish, and brain tissue slices or implanted optical fiber microarrays by recording the fluorescence signal change after electrical stimulation.

Overall, the reported research is novel, innovative and timely, and it should be of significant interest to chemists, life scientists and medical scientists. Unfortunately, the current version still contains several shortcomings, in particular at the very basic compound characterization and purity level as detailed below. The manuscript may become suitable for publication in Nature Communications after all the points below have been properly addressed.

1. Synthesis:

- a) Yields should be given for each reaction.
- b) Compound characterization is insufficient. In all random checks of the ¹³C NMR spectra, the reported no. of signals does not match the expected no. of signals. This includes very simple compounds such as compound 15 (25 C-atoms, 19 signals expected, 29 signals reported), as well as the key compounds G2 (41 C-atoms, 29 signals expected, 45 signals reported) or G3 (43 C-atoms, 31 signals expected, 43 signals reported). In addition, the respective ¹H NMR spectra are not well resolved and contain unassigned peaks (e.g. at ca. 8.5, 6.7, 5.3, 3.0, 1.6, 1.4, and 1.3 ppm for compound G2). Moreover, the mass

spectra are reported only for a very limited range (e.g. at m/z from 680 to 800 for compound G2). In summary, this raises significant concerns about the purity of the investigated compounds. The authors should report clear ^1H NMR and ^{13}C NMR spectra for all compounds using thin line widths to enable inspection of coupling patterns in ^1H NMR and clearly discriminable peaks in ^{13}C NMR. The y-axis of all NMR spectra should be scaled to the compound peaks and not to the much larger solvent residual peaks. HPLC traces of the key compounds are easy to record due to the absorption by the aromatic rings and would eliminate concerns about compound purity.

2. NMR Characterization of host-guest complexes: The authors write at different positions in the manuscript that they have measured NOESY and/or ROESY spectra, which are related (but slightly different) techniques. What has actually been measured should be clarified and the selected mixing times in the NMR pulse programs should be reported. Moreover, the peak assignments may be questionable. It remains unclear, how the authors assigned NMR peaks to the respective hydrogen atoms (e.g. in Figs. 3e and 3f). Without proper ^{13}C NMR analysis (see point above) including 2D NMR spectra, an unequivocal assignment of the peaks appears to be a major challenge.

3. Job's plots and binding stoichiometry: The authors write that they found 1:2 CB[8](CB[10])-G2 complexes (p.5, line 115), which does not match the respective Job's plots in Fig. S65. Moreover, a Job's plot for CB[8] and MB is lacking.

4. Fluorescence titrations: The fluorescence response in Figs. 3b and 3c seems to be different. In Fig. 3b, the fluorescence reaches a plateau of <40% at high EP concentrations, whereas the plateau in Fig. 3c is at ca. 45% of the initial fluorescence intensity. Moreover, the estimation of a binding affinity (Fig. S70 and p. 6, lines 170 and following) is meaningless, since the mechanism of fluorescence detection is irreversible (see below).

5. Mechanisms of fluorescence detection: Since the sensing mechanism involves the irreversible formation of an amide bond from an active ester derivative, the manuscript should be rewritten at several instances. For example, the title (chemosensor) implies a reversibility of the sensing mechanism, whereas irreversible probes are commonly referred to as chemodosimeters. The rapid response of the probes is indeed remarkable, but the authors did not prove that EP dynamics can actually be measured (lines 10, 34, and 55-56), since the covalent bond formation may be rate-limiting. This poses also significant constraints to real-time monitoring (lines 61-62).

6. Fluorescence imaging: The authors used 100 μM EP to demonstrate the fluorescence response in neuronal cells and write that "at the neuron cytomembrane [...] the concentration of EP was 0.07 \pm 0.01 μM (Figure 4g)" (lines 255-259). It remains unclear how the concentration was estimated and whether it compares favorably with literature values. This applies also to the values after stimulation (Figs. 4i and 4j, lines 278-292). It appears unreasonable that the fluorescence ratio changes from ca. 3.8 to 2.8 with 100 μM EP (Figs. 4d and 4e), while a change from ca. 4.0 to 3.4 refers to a EP concentration of ca. 1 μM . This should be clarified.

7. Bioimaging: The authors use electrical stimulation for bioimaging. Since pyridinium and stilbene derivatives (as the guest G2) as well as MB are well-known to be redox-active, control experiments should be performed to exclude the possibility that the fluorescence response does not originate from the fact that these compounds are electrochemically active.

As an additional consideration, I see no benefit of including the DFT calculations in the manuscript. The pK_a values in the range of 40-50 (gas phase values?) are rather confusing than adding value to a manuscript that is concerned about sensing in water.

Reviewer #3

(Remarks to the Author)

A supramolecular fluorescent probe was developed and demonstrated as an EP chemosensor for optical detection applications, particularly in vivo. The design and development process, along with the testing and verification of the molecule, are reported. I believe this is valuable work for neuroscience or basic medical research. Here are a few questions:

1. First of all, the authors should compare the testing results with those of traditional testing methods, especially for experiments conducted on animals.
2. The statement "The millisecond kinetics in neurons lack" requires clarification regarding direct evidence. The first thing that needs to be confirmed is the moment when EP changes significantly within the nervous system.
3. It is generally difficult to perfectly align the stained area with the area illuminated by the optical probe. Although the authors have simulated the calculation of the fiber's illumination area, the situation is more complex in vivo.
4. The authors should explain that the EP concentration increases in some brain regions while remaining unchanged in others. How can they prove that the EP originates from neurons and not from capillaries or extracellular fluid?

Version 1:

Reviewer comments:

Reviewer #1

(Remarks to the Author)

the author addressed all issues, i would like to agree with its publication as this form

Reviewer #2

(Remarks to the Author)

The authors have addressed all my concerns. The manuscript can be published as is.

Reviewer #3

(Remarks to the Author)

I agree with the response and have no more questions

Reply to Reviewer 1:

Recommendation: Minor Revision: suitable for publication after changes

Comments:

The real-time dynamics of epinephrine (EP) is highly relevant to decipher complex brain activities. However, there are currently no probes, including fluorescent protein probes and small molecule probes, that enable rapid detection of EP. To overcome this dilemma, Tian and coworkers reported a first supramolecular fluorescent chemsensor (CMG2) for real-time imaging and quantitative biosensing of EP dynamics in freely behaving animals. The overall performance of CMG2 showed high affinity, sensitivity, and fast kinetics (~ 240 ms) for monitoring EP activity in vitro and in vivo. Importantly, such a chemsensor could achieve real-time monitoring EP concentrations, distributions, and dynamic changes from neuron to whole brain. These data and results are particularly intriguing and critically important in the field of brain science. Overall, this paper can be considered for the publication after minor revision.

Q1. What is the rationale design behind this structure, despite the boronic acid recognition site and active fluorinated phenyl ester? What's the function of piperazine in the guest molecules? Please provide a sentence for the rationale behind this choice.

A1. We greatly appreciate the reviewer's high evaluation and helpful suggestions. In this work, a dual-site functionalized fluorescent guest molecule (G2) was synthesized, which contained a boric acid group at one end and an active fluorinated phenyl ester at the other (Page 3, Line 71-74). The boric acid group is expected to interact with catechol-type neurotransmitters through the condensation reaction; and the active fluorinated phenyl ester can react with monoamine-type neurotransmitters through amidation reaction. Following the complexation of G2 and CB [10] by host-guest interaction, the probe's response speed and selectivity toward EP were improved due to the confinement effect.

The piperazine in G2 serves three main functions. Firstly, its combination with styrylpyridinium salt fluorophores generates an intramolecular charge transfer effect (ICT),^[R1-R3] where electrons transfer from piperazine to the fluorophore backbone, resulting in increased fluorescence intensity and quantum yield. Secondly, the introduction of piperazine unit enhances the planar rigidization of guest molecule through-space conjugation (TSC) effect.^[R4] Finally, the imine group at the opposite end of piperazine acts as a reaction site, facilitating the introduction and attachment of the recognition group. The corresponding issue was added in the revised manuscript (Page 3, Line 74-77).

Q2. The authors should point out the physiological concentrations of EP in the manuscript to demonstrate the applicability of the designed probes.

A2. The concentration of EP - an essential catecholamine neurotransmitter, has been reported from 1.0 to 100 nM [R5-R8]. Various factors can influence EP levels, including the type of neuron, environmental stimuli, and physiological states. For instance, EP levels may temporarily increase due to neural activity or stress response. After electrical stimulation or upon fear, the concentration of EP can increase to 200 nM -1.0 μ M or even higher [R9-R14]. In this work, the linear range for the developed probe **CMG2** to detect EP concentration was from ~20 nM to 12.0 μ M, with a good linear correlation ($R^2 = 0.9943$). The limit of detection (LOD) was estimated to be 5.1 ± 0.3 nM ($n = 5$, S.D.) (Page 6, Line 170-173). Therefore, our probe is well-suited for monitoring EP levels in the complex biological environments.

Q3. There are some errors in chemical structures for guest molecules in SI, e.g., the reagent of di-bromide used to synthesize intermediate 3 should contain not 4.

A3. We are sorry for the mistakes. We corrected 4 methylene groups into 6 methylene groups in the revised Supporting Information (Page S2, Supporting Information).

Q4. What is the scale bar of Figure S80c?

A4. We added the scale bar of 1 cm in the revised Supporting Information (Page S60, Supporting Information).

Reply to Reviewer 2:

Recommendation: Minor Revision: suitable for publication after changes

Comments:

In this manuscript, the authors report self-assembled supramolecular host-guest complexes for the fluorescence-based detection of epinephrine (aka adrenaline). The complexes can be excited around 475 nm and its emission can be detected in the range from ca. 540 nm to ca. 720 nm rendering it suitable for bioimaging. The latter is demonstrated with neuronal cells, zebrafish, and brain tissue slices or implanted optical fiber microarrays by recording the fluorescence signal change after electrical stimulation.

Overall, the reported research is novel, innovative and timely, and it should be of significant interest to chemists, life scientists and medical scientists. Unfortunately, the current version still

contains several shortcomings, in particular at the very basic compound characterization and purity level as detailed below. The manuscript may become suitable for publication in Nature Communications after all the points below have been properly addressed.

Q1. Synthesis: a) Yields should be given for each reaction. b) Compound characterization is insufficient. In all random checks of the ^{13}C NMR spectra, the reported no. of signals does not match the expected no. of signals. This includes very simple compounds such as compound 15 (25 C-atoms, 19 signals expected, 29 signals reported), as well as the key compounds G2 (41 C-atoms, 29 signals expected, 45 signals reported) or G3 (43 C-atoms, 31 signals expected, 43 signals reported). In addition, the respective ^1H NMR spectra are not well resolved and contain unassigned peaks (e.g. at ca. 8.5, 6.7, 5.3, 3.0, 1.6, 1.4, and 1.3 ppm for compound G2). Moreover, the mass spectra are reported only for a very limited range (e.g. at m/z from 680 to 800 for compound G2). In summary, this raises significant concerns about the purity of the investigated compounds. The authors should report clear ^1H NMR and ^{13}C NMR spectra for all compounds using thin line widths to enable inspection of coupling patterns in ^1H NMR and clearly discriminable peaks in ^{13}C NMR. The y-axis of all NMR spectra should be scaled to the compound peaks and not to the much larger solvent residual peaks. HPLC traces of the key compounds are easy to record due to the absorption by the aromatic rings and would eliminate concerns about compound purity.

A1. We greatly appreciate the reviewer's high evaluation and helpful suggestions. We purified the partial products and supplemented the yield for all compounds in the revised Supporting Information. Then, we assigned the peaks in ^1H and ^{13}C NMR spectra accurately of all compounds. Following the reviewer's comments, we have corrected all NMR spectra in the revised Supporting Information (Page S3-43, Supporting Information). The following information is the characterization for the key compounds, including compound 15, G1, G2, and G3 (Figures R1-R8).

Compound 15, ^1H NMR (500 MHz, 298 K, $\text{DMSO}-d_6$) δ 8.94-8.93 (dd, $J_1 = 2.0$ Hz, $J_2 = 5.0$ Hz, 2H), 8.07-8.05 (dd, $J_1 = 2.0$ Hz, $J_2 = 6.5$ Hz, 2H), 8.00-7.98 (d, $J = 7.0$ Hz, 2H), 7.26-7.22 (m, 1H), 7.20-7.18 (dd, $J_1 = 2.5$ Hz, $J_2 = 8.0$ Hz, 2H), 7.12-7.10 (dd, $J_1 = 2.0$ Hz, $J_2 = 7.0$ Hz, 2H), 4.55-4.52 (m, 2H), 4.10-4.08 (m, 2H), 2.61 (s, 3H), 1.96-1.90 (m, 2H), 1.78-1.72 (m, 2H), 1.50-1.44 (m, 2H), 1.36-1.30 (m, 2H). ^{13}C NMR (125 MHz, 298 K, $\text{DMSO}-d_6$) δ 163.52, 163.37, 158.78, 143.71, 135.32, 132.23, 128.36, 125.73, 120.17, 114.73, 106.89, 101.78, 67.87, 59.87, 30.44, 28.17, 25.09, 24.85, 21.36. 25 C-atoms, 19 signals expected, 19 signals reported: ^{13}C NMR (125 MHz, 298 K, $\text{DMSO}-d_6$) δ 163.52, 163.37, 158.78, 143.71, 135.32, 132.23, 128.36,

125.73, 120.17, 114.73, 106.89, 101.78, 67.87, 59.87, 30.44, 28.17, 25.09, 24.85, 21.36;

G1 ¹H NMR (500 MHz, 298 K, DMSO-*d*₆) δ 8.86-8.85 (d, *J* = 6.0 Hz, 2H), 8.13-8.11 (d, *J* = 6.0 Hz, 2H), 8.02 (s, 2H), 7.76-7.75 (d, *J* = 2.5 Hz), 7.61-7.60 (d, *J* = 7.5 Hz, 2H), 7.32-7.29 (m, 4H), 7.24 (s, 1H), 7.13-7.10 (m, 1H), 7.02-7.00 (m, 2H), 6.97-6.96 (dd, *J*₁ = 1.5 Hz, *J*₂ = 5.0 Hz, 2H), 4.89-4.87 (m, 2H), 4.54-4.52 (m, 2H), 4.03-4.01 (m, 2H), 3.53 (s, 2H), 1.99 (s, 2H), 1.60-1.57 (m, 4H), 1.50-1.42 (m, 4H). 39 C-atoms, 27 signals expected, 27 signals reported: ¹³C NMR (125 MHz, 298 K, DMSO-*d*₆) δ 170.34, 168.59, 158.18, 154.06, 152.41, 144.30, 144.08, 141.98, 139.81, 134.10, 130.16, 130.05, 129.40, 128.68, 127.96, 124.58, 122.50, 121.48, 118.45, 114.92, 114.32, 113.32, 59.75, 52.32, 46.82, 30.95, 24.05.

G2, ¹H NMR (500 MHz, 298 K, DMSO-*d*₆) δ 8.85-8.84 (d, *J* = 5.5 Hz, 2H), 8.12-8.11 (d, *J* = 6.0 Hz, 2H), 7.99 (s, 2H), 7.94-7.92 (d, *J* = 13.5 Hz, 2H), 7.76-7.75 (d, *J* = 6.5 Hz, 2H), 7.61-7.60 (d, *J* = 7.5 Hz, 2H), 7.32-7.29 (m, 4H), 7.26-7.24 (d, *J* = 13.0 Hz, 1H), 7.02-7.01 (d, *J* = 7.5 Hz, 2H), 6.96-6.95 (d, *J* = 7.0 Hz, 2H), 4.53-4.51 (m, 2H), 4.06-4.04 (m, 2H), 3.53 (s, 2H), 2.09-2.04 (m, 2H), 1.77-1.72 (m, 4H), 1.62-1.58 (m, 4H), 1.48 (s, 4H). 41 C-atoms, 29 signals expected, 29 signals reported: ¹³C NMR (125 MHz, 298 K, DMSO-*d*₆) δ 168.79, 162.23, 159.15, 153.62, 152.32, 143.72, 141.61, 139.77, 134.10, 131.46, 131.20, 129.96, 128.66, 128.49, 127.96, 126.21, 124.69, 122.80, 118.55, 114.35, 114.12, 113.26, 66.90, 62.01, 58.91, 52.31, 46.85, 27.45, 24.08;

G3, ¹H NMR (500 MHz, 298 K, DMSO-*d*₆) δ 8.83-8.82 (d, *J* = 5.0 Hz, 2H), 8.10-8.09 (d, *J* = 5.5 Hz, 2H), 8.00 (s, 2H), 7.94-7.88 (m, 2H), 7.77-7.75 (d, *J* = 6.5 Hz, 2H), 7.61-7.59 (d, *J* = 7.0 Hz, 2H), 7.30-7.29 (d, *J* = 8.5 Hz, 4H), 7.26 (s, 1H), 7.02-7.00 (d, *J* = 7.5 Hz, 2H), 6.93-6.92 (d, *J* = 7.0 Hz, 2H), 4.46-4.43 (m, 2H), 3.99-3.97 (m, 2H), 3.53 (s, 2H), 1.93-1.90 (m, 2H), 1.73-1.71 (m, 2H), 1.59 (s, 2H), 1.47-1.45 (m, 4H), 1.35-1.32 (m, 4H), 1.24 (s, 2H), 0.87-0.84 (m, 2H). 43 C-atoms, 31 signals expected, 31 signals reported: ¹³C NMR (125 MHz, 298 K, DMSO-*d*₆) δ 168.82, 159.34, 153.56, 152.36, 143.69, 141.59, 134.11, 131.25, 131.11, 131.03, 130.16, 129.95, 128.68, 128.34, 127.96, 122.75, 118.53, 114.93, 114.35, 114.02, 113.34, 113.29, 67.37, 59.25, 52.33, 46.85, 30.96, 30.38, 28.28, 25.14, 24.09).

HPLC-MS analysis of the final products G1, G2, and G3 was supplemented to confirm their purity. The analysis was performed on 1200HPLC+microTOF II using a Sharpsil-u C18 column (analytical column, 250 mm × 21.2 mm, particle size 5 μm) at 25°C. Isocratic elution was applied using 100 % mobile phase A (methanol) and 0 % mobile phase B (water) at 25 °C. The flow rate was set at 1 mL/min, and the injection volume was 50 μL. Detection was performed at 254 nm. As shown in Figures R9 and Table R1, the target peak area corresponding to G1 was observed at a mass-to-charge ratio (*m/z*) of 676.269 (*t_r* = 2.98 min), for G2 at *m/z* 704.306 (*t_r* =

2.94 min), and for G3 at m/z 732.342 ($t_r = 2.77$ min). All compounds exhibited high purity (>95%) [R15-R19]. Figures R1-R9 and Table R1 were added as Figures S43-44, S55-56, S58-59, S61-62, S64 and Table S1 in the revised Supporting Information (Page S30-43, Supporting Information). The corresponding issue was also added in the revised manuscript (Page 3, Line 82-85).

Figure R1 ^1H NMR (500 MHz, 298 K, DMSO-d_6) spectrum of Compound 15.

Figure R2. ¹³C NMR (125 MHz, 298 K, DMSO-d₆) spectrum of compound 15.

Figure R3. ¹H NMR (500 MHz, 298 K, DMSO-d₆) spectrum of G1.

Figure R4. ^{13}C NMR (125 MHz, 298 K, $\text{DMSO}-d_6$) spectrum of G1.

Figure R5. ^1H NMR (500 MHz, 298 K, $\text{DMSO}-d_6$) spectrum of G2.

Figure R6. ¹³C NMR (125 MHz, 298 K, DMSO-d₆) spectrum of G2.

Figure R7. ¹H NMR (500 MHz, 298 K, DMSO-d₆) spectrum of G3.

Figure R8. ^{13}C NMR (125 MHz, 298 K, $\text{DMSO}-d_6$) spectrum of G3.

Figure R9. HPLC-MS spectra of (a) G1, (b) G2, and (c) G3.

Table R1. HPLC-MS results of G1, G2 and G3.

Compound	Retention time (min)	Area (nAU*min)	%Area
G1	2.9801	6516.4527	95.01
G2	2.9467	23819.1264	96.69
G3	2.7764	30550.1641	95.11

Q2. NMR Characterization of host-guest complexes: The authors write at different positions in the manuscript that they have measured NOESY and/or ROESY spectra, which are related (but slightly different) techniques. What has actually been measured should be clarified and the

selected mixing times in the NMR pulse programs should be reported. Moreover, the peak assignments may be questionable. It remains unclear, how the authors assigned NMR peaks to the respective hydrogen atoms (e.g. in Figs. 3e and 3f). Without proper ^{13}C NMR analysis (see point above) including 2D NMR spectra an unequivocal assignment of the peaks appears to be a major challenge.

A2. We greatly appreciate the reviewer's thoughtful review and suggestions. In this work, we performed ROESY, not NOESY, to characterize the host-guest interactions, and we have corrected the terminology in the revised manuscript (Page 8, Line 211). While NOESY and ROESY are related techniques, NOESY requires a pre-set mixing time to induce the Nuclear Overhauser Effect, whereas ROESY does not. Of note, all necessary ^1H and ^{13}C NMR have been recorded properly. To obtain the detailed structural characterization of the supramolecular complex **CMG2**, we performed 2D ^1H - ^1H COSY NMR for the guest molecule G2. As shown in Figure R10, the signals for H₁, H₂, H₃, H₄, H₅, H₆, H₁₁, and H₁₂ were assigned at 8.84, 8.11, 7.29, 7.99, 6.95, 7.31, 7.60, and 7.61 ppm, respectively. These unequivocal assignments allowed us to investigate the self-assembly behavior through NMR experiments. Figure R10 was added as Figure S77 in the revised Supporting Information (Page S54, Supporting Information). The corresponding issue was also added in the revised manuscript (Page 8, Line 202-205).

Figure R10. 2D COSY spectrum of the guest molecule G2.

Q3. Job's plots and binding stoichiometry: The authors write that they found 1:2 CB[8](CB[10])-G2 complexes (p.5, line 115), which does not match the respective Job's plots in Fig. S65. Moreover, a Job's plot for CB[8] and MB is lacking.

A3. We agreed with the reviewer. We supplemented the job's titration experiments for CB[8]-MB, CB[10]-MB, CB[8]-G2 and CB[10]-G2. As shown in Figure R11, the corresponding binding

ratios for CB[8]-MB, CB[10]-MB, CB[8]-G2 and CB[10]-G2 were determined to be 1:2, 1:2, 1:2 and 1:1, respectively, which aligns well with previous reports^[R20,R21]. Figure R11 was added as Figure S66 in the revised Supporting Information (Page S46). The corresponding issue was also added in the revised manuscript (Page 5, Line 118-120).

Figure R11. Fluorescence spectra of MB+CB[8] (a), MB+CB[10] (c), G2 + CB[8] (e), and G2 + CB[10] (g) in aqueous solution with difference host : guest ratios; Job's plot for the complexation between MB and CB[8] (b) (at 695 nm), MB and CB[10] (d) (at 695 nm), G2 and CB[8] (f) (at 570 nm) and G2 and CB[10] (h) (at 570 nm) (the total concentration was fixed as 20 μM). (Error bars, n = 5, S.D.).

Q4. Fluorescence titrations: The fluorescence response in Figs. 3b and 3c seems to be different. In Fig. 3b, the fluorescence reaches a plateau of < 40% at high EP concentrations, whereas the plateau in Fig. 3c is at ca. 45% of the initial fluorescence intensity. Moreover, the estimation of a binding affinity (Fig. S70 and p. 6, lines 170 and following) is meaningless, since the mechanism of fluorescence detection is irreversible (see below).

A4. We greatly appreciate the reviewer's suggestion and evaluation. To address this issue, we performed five repeated experiments (Figures R12a-e). The changes in fluorescence intensity of F₅₇₀ and F₆₉₅ with increasing EP concentration are shown in Figure R12f. When the EP concentration reached 12 μM, the plateau values of F₅₇₀ channel were 0.410, 0.428, 0.434, 0.450 and 0.460, with an average of 0.436 ± 0.0195 (Table R2). For the F₆₉₅ channel, the plateau intensities were 0.210, 0.230, 0.230, 0.225 and 0.225, yielding an average value of 0.224 ± 0.0082. The normalized F₅₇₀/F₆₉₅ ratio at plateau was 0.437 ± 0.0253, consistent with the single-channel measurements. By analyzing the average fluorescence intensities of both channels at varying EP concentrations, we calculated the F₅₇₀/F₆₉₅ ratio and established a linear relationship between the fluorescence ratio and EP concentration (0.02-12 μM), as shown in Figures R12g-h.

The error bars in the above statistical results represent the standard deviation (S.D.), and the formula for calculating the S.D. is as follows:

$$S. D. = \sqrt{\frac{\sum_{i=1}^n (x_i - \bar{x})^2}{n - 1}}$$

$$S. D. (z) = \sqrt{\left(\frac{SD_x}{y}\right)^2 + \left(\frac{xSD_y}{y^2}\right)^2}$$

Figure R12 was added as Figure S72 in the revised Supporting Information (Page S52). The corresponding issue was also added in the revised manuscript (Page 6, Line 167-170).

In this work, the binding affinity between **CMG2** and EP was calculated to be $4.66 \times 10^4 \text{ M}^{-1}$ (Page 6, Line 173-175). Although the fluorescence detection mechanism is generally considered irreversible, the binding energy is intrinsically linked to the selectivity of the probe toward EP. In the research of host-guest complexes and biochemical investigations, binding constant is commonly employed to assess the affinity between a molecule and its target, reflecting the stability of the complex [R22-R24]. The binding constant of our probe with EP ($4.66 \times 10^4 \text{ M}^{-1}$) demonstrates a strong affinity between them, confirming high selectivity of our probe toward EP.

Table R2. Fluorescence intensity of single and dual channels when 12.0 μM of EP was added.

	1	2	3	4	5	Mean	S.D.
F ₅₇₀	0.41	0.428	0.434	0.45	0.46	0.436	0.0195
F ₆₉₅	0.21	0.23	0.23	0.225	0.225	0.224	0.0082
F _{570/695}	/	/	/	/	/	1.946	0.1125
F _{570/695} (normalization)	/	/	/	/	/	0.437	0.0253

Figure R12. (a-e) Fluorescence spectra of 10.0 μM **CMG2** with the addition of EP at different concentrations (0- 15 μM) in cell lysis buffer (10 mM, pH = 7.4) containing 0.05% DMSO excited at 480 nm. (f) Single channel F_{570} (black) and F_{695} (red), (g) F_{570}/F_{695} ratio and (h) Relative fluorescence intensity (normalization) of the **CMG2** versus EP concentration (0- 15 μM) obtained from a-e (error bars, n = 5, S.D.).

Q5. Mechanisms of fluorescence detection: Since the sensing mechanism involves the irreversible formation of an amide bond from an active ester derivative, the manuscript should be rewritten at several instances. For example, the title (chemosensor) implies a reversibility of the sensing mechanism, whereas irreversible probes are commonly referred to as chemodosimeters. The rapid response of the probes is indeed remarkable, but the authors did not prove that EP dynamics can actually be measured (lines 10, 34, and 55-56), since the covalent bond formation may be rate-limiting. This poses also significant constraints to real-time monitoring (lines 61-62).

A5. We replaced the word “chemosensors” with “chemodosimeters” in the revised manuscript accordingly (Page 1, Line 2). In this work, we utilized **CMG2** to monitor EP dynamics in vitro with a temporal resolution of approximately 240 ms (Page 5-6, Line 143-144). The fast fluorescence response of the probes toward EP was achieved using a rapid-mixing stopped-flow technique, employing a pneumatic drive system (SFA-20, HI-Tech, TgK Scientific, United

Kingdom), combined with a fluorescence spectrometer (Hitachi F-4600, Japan), as shown in Figure R14. Equal volumes of the probe solution ($10 \mu\text{M}$, $150 \mu\text{L}$, syringe A) and pure water ($150 \mu\text{L}$, syringe B) were quickly mixed in a highly efficient mixer while monitoring the basal fluorescence intensity at 570 nm . Subsequently, the probe ($20 \mu\text{M}$, $150 \mu\text{L}$, syringe A) and EP ($10 \mu\text{M}$, $150 \mu\text{L}$, syringe B') solutions were rapidly combined in the same manner to initiate the reaction. This mixing process displaced the contents of the optical cell ($10 \mu\text{M}$, $300 \mu\text{L}$ of the probe solution), filling it with freshly mixed reagents, while continuously monitoring fluorescence intensity. The total injected volume was controlled by a stop syringe, providing the “stopped-flow” capability. The dead time of such mixing system was ca. 8 ms , and the fluorometer has a shorter sampling interval (5 ms).

EP dynamics were effectively measured by our rapid-mixing stopped-flow technique. Previous studies have reported that the covalent bond formation occurs very rapidly. For instance, time-resolved infrared spectroscopy confirmed that the formation of covalent bonds between cysteine and flavin is completed within 3 ms [R25, R26]. In addition, in our previous work, we successfully achieved ultra-fast detection of norepinephrine on a 100-ms timescale using the rapid-mixing stopped-flow technique [R27, R28], reinforcing the feasibility of our measurements within this time range. Figure R13 was added as Figure S69 in the revised Supporting Information (Page S48, Supporting Information). The corresponding issue was also added in the revised manuscript (Page 5, Line 133-135).

Figure R13. (a) Picture of the stopped-flow accessory with a pneumatic drive system; (b) Schematic of the stopped-flow-based kinetics measurement.

Q6. Fluorescence imaging: The authors used $100 \mu\text{M}$ EP to demonstrate the fluorescence response in neuronal cells and write that "at the neuron cytomembrane [...] the concentration of EP was $0.07 \pm 0.01 \mu\text{M}$ (Figure 4g)" (lines 255-259). It remains unclear how the

concentration was estimated and whether it compares favorably with literature values. This applies also to the values after stimulation (Figs. 4i and 4j, lines 278-292). It appears unreasonable that the fluorescence ratio changes from ca. 3.8 to 2.8 with 100 μM EP (Figs. 4d and 4e), while a change from ca. 4.0 to 3.4 refers to a EP concentration of ca. 1 μM. This should be clarified.

A6. We greatly appreciate the reviewer's suggestion and evaluation. Upon further examination, we identified the mistake in the calculation process of concentration quantification. We sincerely apologize for this mistake.

We initially analyzed the F_{570}/F_{695} ratio on resting neuronal cell membranes (3.95 ± 0.014). Using the established linear relationship between EP concentration and the F_{570}/F_{695} ratio from neuron lysates ($F_{570}/F_{695} = 3.965 - 0.166 [\text{EP}] \mu\text{M}$) (Figure R14), we calculated the EP concentration on the membrane to be $0.09 \pm 0.02 \mu\text{M}$, consistent with literature values (reported from 1.0 to 100 nM) ^[R5-R8] (Figures 4i and 4j, Page 11, Line 299-301). Subsequently, we applied 100 μM EP, a saturated concentration, to assess the fluorescence response in neuronal cells ^[R29, R30] (Figures 4d and 4e, Page 9-10, Line 259-260 and Line 277). This resulted in a decrease in the F_{570}/F_{695} ratio from 3.95 ± 0.022 to 2.77 ± 0.10 , indicating an increase in EP concentration from $0.09 \pm 0.03 \mu\text{M}$ to $7.20 \pm 0.16 \mu\text{M}$. We hypothesized that the high concentration of exogenous EP might cause receptor saturation or downregulation, diminishing further EP uptake by neurons ^[R31-R33]. To confirm this hypothesis, we supplemented the fluorescence response by adding varying concentrations of EP. As shown in Figure R15, the application of 1 μM EP decreased the F_{570}/F_{695} ratio of **CMG2** from 3.95 ± 0.013 to 3.80 ± 0.024 (EP concentration from $0.09 \pm 0.02 \mu\text{M}$ to $0.99 \pm 0.04 \mu\text{M}$); while 5 μM EP reduced it further to 3.14 ± 0.047 (EP from $0.09 \pm 0.02 \mu\text{M}$ to $4.97 \pm 0.07 \mu\text{M}$), and 10 μM EP decreased the ratio to 2.76 ± 0.085 (EP from $0.09 \pm 0.03 \mu\text{M}$ to $7.20 \pm 0.13 \mu\text{M}$). Notably, we observed no further change in the F_{570}/F_{695} ratio when the concentration of exogenous EP reached 10 μM, suggesting that neuronal uptake of EP saturates at about $7.20 \pm 0.13 \mu\text{M}$.

Regarding the dynamic release of EP from neuronal cell membranes during electrical stimulation (20 Hz, 3 V, sinusoidal waveform for 1.0 s), we observed a decrease of F_{570}/F_{695} ratio from 3.95 ± 0.014 to 3.44 ± 0.042 , corresponding to an increase in EP concentration from $0.09 \pm 0.02 \mu\text{M}$ to $3.16 \pm 0.07 \mu\text{M}$, as determined by the linear equation ($F_{570}/F_{695} = 3.965 - 0.166 [\text{EP}] \mu\text{M}$) (Figures 4i and 4j, Page 11 Line 299-301).

Figures R14 and R15 were added as Figures S73 and S82 in the revised Supporting Information (Page S52 and S57). The corresponding issue was also added in the revised manuscript (Page 6, Line 171-172 and Page 11, Line 299-301).

Figure R14. The plot of F_{570}/F_{695} ratio (F_{570} : 505 - 645 nm, F_{695} : 665 - 750 nm) of the **CMG2** versus EP concentration (error bars, $n = 5$, S.D.).

Figure R15. Representative traces of **CMG2** in response to (a) 1 μM EP; (c) 5 μM EP; (e) 10 μM EP; Dynamic response summary of **CMG2** in response to (b) 1 μM EP (EP from $0.09 \pm 0.02 \mu\text{M}$ to $0.99 \pm 0.04 \mu\text{M}$); (d) 5 μM EP (EP from $0.09 \pm 0.02 \mu\text{M}$ to $4.97 \pm 0.07 \mu\text{M}$); (f) 10 μM EP (EP from $0.09 \pm 0.03 \mu\text{M}$ to $7.20 \pm 0.13 \mu\text{M}$). The fluorescence intensity error was represented by the standard deviation (SD), while the concentration error was calculated using the standard error of the mean (SEM). Red dots represent individual data points. Statistical significance was calculated with a two-tailed unpaired t-test ($n = 15$; *** $p < 0.001$).

Q7. Bioimaging: The authors use electrical stimulation for bioimaging. Since pyridinium and stilbene derivatives (as the guest G2) as well as MB are well-known to be redox-active, control experiments should be performed to exclude the possibility that the fluorescence response does not originate from the fact that these compounds are electrochemically active.

A7. We appreciate the constructive suggestion provided by the reviewer. In this work, we employed electrical stimulation (20 Hz, 3 V, sine wave for 1.0 s) as the most direct and rapid approach to trigger neuronal action potential, using a microelectrode approaching the outside of neuron membranes. To address the concern regarding the potential electrochemical activity of these compounds, we conducted control experiments to ensure that the observed fluorescence response was not a result of such activity. Specifically, we applied an external voltage (20 Hz, 3 V, sine wave for 1.0 s) to the **CMG2** solution and monitored changes in fluorescence intensity. As shown in Figure R16a-16b, the fluorescence intensity of **CMG2** remained essentially unchanged after electrical stimulation. Furthermore, we performed NMR and MALDI mass spectrometry analysis of the **CMG2** solution after applying the voltage and found no structural changes (Figure R16c-16d). These results collectively confirm that the fluorescence response is not attributed to the redox activity of the prob **CMG2**. Figure R16 was added as Figure S83 in the revised Supporting Information (Page S57, Supporting Information). The corresponding issue was also added in the revised manuscript (Page 10-11, Line 278-279 and 293-296).

Figure R16. (a) Fluorescent intensity of **CMG2** in solution after electrical stimulation (20 Hz, 3 V, sine wave for 1.0 s); (b) Dynamic responses of **CMG2** in solution after electrical stimulation. The data were presented as mean \pm S.D. Error bars: S.D., red dots represent individual data points. Statistical significance was calculated with a two-tailed unpaired t-test ($n = 15$; $^{ns}p > 0.05$); (c) ¹H NMR spectra of **CMG2** after electrical stimulation (20 Hz, 3 V, sine wave for 1.0 s) (i) and **CMG2** (ii); (d) Maldi-TOF mass spectrometry of **CMG2** after electrical stimulation (20 Hz, 3 V, sine wave for 1.0 s).

Q8. As an additional consideration, I see no benefit of including the DFT calculations in the manuscript. The pKa values in the range of 40-50 (gas phase values?) are rather confusing than adding value to a manuscript that is concerned about sensing in water.

A8. Thank you for your constructive comments on the manuscript. In this work, to further explain the high selectivity of the probe toward EP, we conducted first-principle calculation based on Density Functional Theory (DFT). Of note, pKa of EP, DA, and NE were 45.5749, 42.4725, and 41.9999, respectively. Electrostatic Potential Energy (ESP) distribution and Natural Population Analysis (NPA) charge calculation showed that the electronegativity of the nitrogen atoms in EP, DA and NE is -0.653 (EP) < -0.641 (DA) < -0.636 (NE) (Page 6, Line 147-153). These findings suggest that EP exhibits greater reactivity toward the active fluorinated phenyl ester site of **CMG2** compared to NE and DA.

We further calculated pKa in the solution phase using a thermodynamic combination method, with the specific formulas as shown in Equations (1) and (2). The Gibbs free energy change in the solution phase ($\Delta G_{\text{aq}}^{1\text{M}}$) was derived from the gas phase free energy, with proton solvation free energy estimated from literature [R34]. The free energy change from the gas phase standard state to the solution phase standard state ($\Delta G^{1\text{atm}\rightarrow 1\text{M}}$) was determined to be 1.89 kcal/mol. The calculation formula for the free energy change in the solution phase was shown in (3) [R35, R36]. By comparing the EP, NE and DA pKa values, it was found that the EP with the highest pKa value is more likely to react with the fluorophenyl ester moiety of **CMG2**. The corresponding issue was also added in the revised Supporting Information (Page S51).

$$\text{AH}_{\text{aq}} = \text{A}_{\text{aq}}^- + \text{H}_{\text{aq}}^+ \quad (1)$$

$$\text{pK}_a = -\log_{10}(K_a) = \frac{\Delta G_{\text{aq}}}{2.303RT} \quad (2)$$

$$\Delta G_{\text{aq}}^{1\text{M}} = G_{\text{gas}}^0(\text{A}^-) + G_{\text{gas}}^0(\text{H}^+) - G_{\text{gas}}^0(\text{AH}) + \Delta G^{1\text{atm}\rightarrow 1\text{M}} + \Delta G_{\text{solv}}^{\text{mod}}(\text{A}^-) + \Delta G_{\text{solv}}^{\text{mod}}(\text{H}^+) - \Delta G_{\text{solv}}^{\text{mod}}(\text{AH}) \quad (3)$$

Reply to Reviewer 3:

Recommendation: Minor Revision: suitable for publication after changes

Comments:

A supramolecular fluorescent probe was developed and demonstrated as an EP chemosensor for optical detection applications, particularly in vivo. The design and development process, along with the testing and verification of the molecule, are reported. I believe this is valuable work for neuroscience or basic medical research. Here are a few questions:

Q1. First of all, the authors should compare the testing results with those of traditional testing methods, especially for experiments conducted on animals.

A1. We greatly appreciate the reviewer's high evaluation and helpful suggestions. We summarized the analytical performance for detecting EP, including linear ranges and limits of detection obtained from traditional mass spectrometry, liquid chromatography, electrochemical methods, fluorescence methods, etc., and compared these with our current method (Table R3). In comparison, our method achieved the lowest limit of detection (LOD = 5.1 ± 0.3 nM). However, the aforementioned reports (mass spectrometry, liquid chromatography, electrochemical, fluorescence methods, etc) [R37-R43] were unable to detect EP in cells or in vivo. Our method enables real-time imaging and quantification of EP with high specificity and rapid kinetics in neurons after electrical stimulation. A recent study utilized an NIR fluorescent probe for EP detection in living HeLa cells, with a detection linear range from 2 μ M to 75 μ M and an LOD of 0.4 μ M [R44]. In stark contrast, the linear range for the developed probe **CMG2** for detecting EP was approximately 20 nM to 12.0 μ M, which is significantly lower than the minimum detection range of the method using the NIR fluorescence probe. Additionally, the LOD of 5.1 ± 0.3 nM is lower than that obtained by the NIR fluorescent probe (Page 6, Line 170-173). Moreover, the developed probe **CMG2** was successfully used for imaging EP in brain tissues and zebrafish, and we established EP-associated molecular networks in 26 regions in deep brain of freely behaving mice for the first time.

Table R3. Comparison of various methods reported for EP determination with our developed method.

Method	Material	Linear concentration range (μ M)	LOD (μ M)	Model	Ref
Electrochemical	SnO ₂ -Graphene	0.5-200	0.017	In vitro	[R37]
Electrochemical	Graphene nanoribbons	6.4-100	2.1	In vitro	[R38]
Electrochemical	Ordered mesoporous carbon/nickel oxide	0.8-50	0.085	In vitro	[R39]
Liquid chromatography	Microdialysis probe	4.09-546	2.73	In vitro	[R40]
-tandem mass spectrometry	Microdialysis probe	2.73-546	0.818	In vitro	[R41]

chromatography					
-tandem mass spectrometry					
Optical fiber biosensor	Aminosilane-functionalized graphene oxide	0.3–90	0.070	In vitro	[R42]
Fluorescence	Ethylene diamine (EDA)	0-10	0.0245	In vitro	[R43]
Fluorescence	Anionic heptamethine cyanine probe	2-75	0.4	In cells	[R44]
Fluorescence	The present method	0.02-12	0.0051	In cells, tissues, zebrafish and mice	--

Q2. The statement "The millisecond kinetics in neurons lack" requires clarification regarding direct evidence. The first thing that needs to be confirmed is the moment when EP changes significantly within the nervous system.

A2. Thank you for your thoughtful comment. Epinephrine levels in the nervous system exhibit marked fluctuations during critical events such as stress responses, fright, or fear ^[R45-R49]. In this work, we measured **CMG2** for monitoring of EP dynamics on ~240 ms in vitro (Page 5-6, Line 143-144). Due to the rapid kinetic response of **CMG2** to EP, it was employed in neurons, brain slices, and zebrafish to assess EP dynamics following electrical stimulation. We also achieved real-time monitoring of EP across 26 deep brain regions in mice, highlighting elevated EP levels in the amygdala, thalamus, hypothalamus, hippocampus, and striatum under fear-induced stress. Following the reviewer's comment, we have corrected "millisecond kinetics" into "rapid kinetics" in the revised manuscript. The corresponding issue was added in the revised manuscript (Page 1, Line 15).

Q3. It is generally difficult to perfectly align the stained area with the area illuminated by the optical probe. Although the authors have simulated the calculation of the fiber's illumination area, the situation is more complex in vivo.

A3. To enhance the reliability of our data, we employed a ratio-metric fluorescent probe to mitigate the effects of probe concentration, light sources, and measurement environments, ensuring the detection accuracy. Additionally, we utilized fiber optic simulation to analyze light field distribution, allowing the fiber to capture the average signal from multiple points and provide a representative value for the entire sample. In our follow-up work, we plan to construct a high-density fiber optic array with additional fibers to capture data across entire brain regions, minimizing regional errors associated with probe illumination.

Q4. The authors should explain that the EP concentration increases in some brain regions while remaining unchanged in others. How can they prove that the EP originates from neurons and not from capillaries or extracellular fluid?

A4. We appreciate the reviewer's insightful comments. In this study, we measured the EP concentrations across seven brain regions and found that fear-induced stress significantly increased EP levels in amygdala, thalamus, hypothalamus, hippocampus, and striatum (part of the basal ganglia), while no significant changes were observed in the cortex and lateral ventricle. The brain regions showing significant changes corresponded to nine sub-regions, including the basolateral amygdaloid nucleus (BLA), basomedial amygdaloid nucleus (BMA), posterior nucleus of thalamus (PO), ventrolateral thalamic nucleus (VL), ventral posteromedial nucleus (VPM), lateral habenular nucleus (LHb), lateral hypothalamus (PLH), field CA3 of the hippocampus (CA3), and caudate putamen (CPu). These changes can be attributed to two main factors. Firstly, these affected brain regions are rich in adrenergic receptors, allowing them to respond to EP released by the locus coeruleus [R50, R51]. Secondly, these regions play crucial roles in regulating emotions, stress responses, and memory formation. The amygdala is particularly involved in emotional behaviors, especially anxiety and conditioned fear, while the hippocampus enhances the retention of fear memories [R51-R54]. The paraventricular thalamic nucleus, sensitive to stress, regulates fear processing in the amygdala [R55-R58]. In contrast, other areas, like the cortex, are less responsive to adrenergic responses and rely more on norepinephrine, dopamine, and GABA for regulation [R59-R61].

To confirm that the responses toward EP were originated from neuronal sources rather than from capillaries or extracellular fluid, we performed fluorescence imaging of brain tissue slices from primary somatosensory cortex (S1BF) and caudate putamen (CPu) regions. As shown in Figure R17, the co-staining brain tissue slices with **CMG2** and a commercial membrane dye Dio for 30 min showed a strong overlay between the fluorescence signals of F₅₇₀ channel of **CMG2** and those of Dio, with a high Pearson correlation coefficient of 0.95,

indicating excellent cytomembrane targeting ability of probe **CMG2**. Figure R17 was added as Figure S91 in the revised Supporting Information (Page S64, Supporting Information). The corresponding issue was also added in the revised manuscript (Page 14, Line 359-362).

Figure R17. Confocal fluorescence images of brain tissue slices in S1BF costained with **CMG2** and a commercial membrane probe (Dio); Scale bar: 50 μ m. (b) Fluorescence images of brain tissue slices from different subregions; Scale bar: 50 μ m.

REFERENCES

- [R1] Yan, L., Su, C., Shen, L., Lv, M., Liu, C., Liu, X., Liu, G., Li, J., Ye, Z. The design and properties study of a novel styryl-pyridinium-based water-soluble fluorescent copolymer as tracing agent. *J. Appl. Polym. Sci.* **136**, 47062 (2019).
- [R2] Zhang, G., Zhang, X., Kong, L., Wang, S., Tian, Y., Tao, X., & Yang, J. Anion-controlled dimer distance induced unique solid-state fluorescence of cyano substituted styrene pyridinium. *Sci. Rep.* **6**, 37609 (2016).
- [R3] Zhang, Y., Li, H., Mai, H., Luo, D., Ji, X., Liu, Z., Peng, S., Xu, X., Zhang, Y., Lan, R., Li, H. A responsive fluorescent probe for detecting and imaging pyruvate kinase M2 in live cells. *Chem. Commun.* **58**, 6494-6497 (2022).
- [R4] Du, Y., Liu, Y., Li, J., He, Y., Li, Y., & Yan, H. Nonconventional Luminescent Piperazine-Containing Hyperbranched Polysiloxanes with Pure n-electron. *Small* **19**, 112809 (2023).
- [R5] Bergquist, J., Sciubisz, A., Kaczor, A., & Silberring, J. Catecholamines and methods for their identification and quantitation in biological tissues and fluids. *J. Neurosci. Meth.* **113**, 1-13. (2002).
- [R6] Lu, X., Li, Y., Du, J., Zhou, X., Xue, Z., Liu, X., & Wang, Z. A novel nanocomposites sensor for epinephrine detection in the presence of uric acids and ascorbic acids. *Electrochim. Acta* **56**, 7261-7266 (2011).
- [R7] Tran Duy, T., Balamurugan, J., Nguyen Thanh, T., Jeong, H., Lee, S. H., Kim, N. H., & Lee, J. H. Enhanced electrocatalytic performance of an ultrafine AuPt nanoalloy framework embedded in graphene towards epinephrine sensing. *Biosens. Bioelectron.* **89**, 750-757 (2017).
- [R8] Anithaa, A. C., Asokan, K., & Sekar, C. Voltammetric determination of epinephrine and xanthine based on sodium dodecyl sulphate assisted tungsten trioxide nanoparticles. *Electrochim. Acta* **237**, 44-53 (2017).
- [R9] Farrell, D. M., Wei, C. C., Tallaj, J., Ardell, J. L., Armour, J. A., Hageman, G. R., Bradley, W. E., Dell'Italia, L. J. Angiotensin II modulates catecholamine release into interstitial fluid of canine myocardium in vivo. *Am. J. Physiol-heart.* **281**, H813-H822 (2001).
- [R10] Tsuda, K., Tsuda, S., & Masuyama, Y. Enhanced endogenous epinephrine release from the vascular adrenergic neurons in spontaneously hypertensive rats. *Am. J. Hypertens.* **3**, 52-54 (1990).
- [R11] Underwood, M. D., Iadecola, C., Sved, A., & Reis, D. J. Stimulation of CI area neurons globally increases regional cerebral blood flow but not metabolism. *J. Cerebr. Blood. F. Met.* **12**, 844-855 (1992).
- [R12] Coco, M. L., Kuhn, C. M., Ely, T. D., & Kilts, C. D. Selective activation of mesoamygdaloid dopamine neurons by conditioned stress: attenuation by diazepam. *Brain Res.* **590**, 39-47 (1992).
- [R13] Sousa, V. C., Mantas, I., Stroth, N., Hager, T., Pereira, M., Jiang, H., Jabre, S., Paslawski, W., Stiedl, O., Svenningsson, P. P11 deficiency increases stress reactivity along with HPA axis and autonomic hyperresponsiveness. *Mol. Psychiatr.* **26**, 3253-3265 (2021).
- [R14] Wimalawansa, S. J. Mechanisms of Developing Post-Traumatic Stress Disorder: New Targets for Drug Development and Other Potential Interventions. *Cns. Neurol. Disord-Dr.* **13**, 807-816 (2014).
- [R15] Ahluwalia, V. K. High Performance Liquid Chromatography (HPLC). In V. K. Ahluwalia (Ed.), *Instrumental Methods of Chemical Analysis* 55-57 (2023).
- [R16] Zhang, C., Mamattursun, A., Ma, X., Pang, T., Wu, Y., & Ma, X. High-performance thin-layer chromatography and high-performance liquid chromatography determination of two anthocyanins in medicine mulberry. *J. Planar Chromatogr. Mod. TLC* **37**, 345-355 (2024).
- [R17] Qin, Y., Li, S., & Zhao, J. HPLC and HPLC-MS for Qualitative and Quantitative Analysis of Chinese Medicines. In S. Li & J. Zhao (Eds.), *Quality Control of Chinese Medicines: Strategies and Methods* 475-577 (2024).
- [R18] Malhotra, P. High Performance Liquid Chromatography. In P. Malhotra (Ed.), *Analytical Chemistry: Basic Techniques and Methods* 263-281 (2023).
- [R19] Loibl, S. F., Harpaz, Z., Zitterbart, R., & Seitz, O. Total chemical synthesis of proteins without HPLC purification. *Chem. Sci.* **7**, 6753-6759. (2016).
- [R20] Montes-Navajas, P., Corma, A., & Garcia, H. Complexation and fluorescence of tricyclic basic dyes encapsulated

- in cucurbiturils. *Chemphyschem* **9**, 713-720 (2008).
- [R21] Fuenzalida, T., & Fuentealba, D. A study of the Fenton-mediated oxidation of methylene blue-cucurbit[*n*] ural complexes. *Photoch. Photobio. Sci.* **14**, 686-692 (2015).
- [R22] Appel, E. A., Biedermann, F., Hoogland, D., del Barrio, J., Driscoll, M. D., Hay, S., Wales, D. J., Scherman, O. A. Decoupled Associative and Dissociative Processes in Strong yet Highly Dynamic Host-Guest Complexes. *J. Am. Chem. Soc.* **139**, 12985-12993 (2017).
- [R23] Schneider, H.-J., & Yatsimirsky, A. K. Selectivity in supramolecular host-guest complexes. *Chem. Soc. Rev.* **37**, 263-277 (2008).
- [R24] Panchal, M., Kongor, A., Athar, M., Modi, K., Patel, C., Dey, S., Vora, M., Bhadresha, K., Rawal, R., Jha, P. C., Jain, V. K. Structural motifs of oxacalix[4]arene for molecular recognition of nitroaromatic explosives: Experimental and computational investigations of host-guest complexes. *J. Mol. Liq.* **306**, 112809 (2020).
- [R25] Konold, P. E., Mathes, T., Weissenhorn, J., Groot, M. L., Hegemann, P., & Kennis, J. T. M. Unfolding of the C-Terminal J α Helix in the LOV₂ Photoreceptor Domain Observed by Time-Resolved Vibrational Spectroscopy. *J. Phys. Chem. Lett.* **7**, 3472-3476 (2016).
- [R26] Pfeifer, A., Majerus, T., Zikihara, K., Matsuoka, D., Tokutomi, S., Heberle, J., & Kottke, T. Time-Resolved Fourier Transform Infrared Study on Photoadduct Formation and Secondary Structural Changes within the Phototropin LOV Domain. *Biophys. J.* **96**, 1462-1470 (2009).
- [R27] Mao, L., Han, Y., Zhang, Q.-W. & Tian, Y. Two-photon fluorescence imaging and specifically biosensing of norepinephrine on a 100-ms timescale. *Nat. Commun.* **14**, 1419 (2023).
- [R28] Han, Y., Mao, L., Zhang, Q.-W., & Tian, Y. Sub-100 ms Level Ultrafast Detection and Near-Infrared Ratiometric Fluorescence Imaging of Norepinephrine in Live Neurons and Brains. *J. Am. Chem. Soc.* **2023**, *145*, 23832-23841.
- [R29] Feng, J., Zhang, C., and Lischinsky, J. E., et al. A Genetically Encoded Fluorescent Sensor for Rapid and Specific *In Vivo* Detection of Norepinephrine. *Neuron* **102**, 745-761 (2019).
- [R30] Wan, J., Peng, W., and Li, X., Qian, et al. A genetically encoded sensor for measuring serotonin dynamics. *Nat. Neurosci.* **24**, 746-752 (2021).
- [R31] Gebauer, L., Rafehi, M., & Brockmoeller, J. Stereoselectivity in the Membrane Transport of Phenylethylamine Derivatives by Human Monoamine Transporters and Organic Cation Transporters 1, 2, and 3. *Biomolecules* **12**, 1507 (2022).
- [R32] Iversen, L. L. Uptake Processes for Biogenic Amines. In L. L. Iversen, S. D. Iversen, & S. H. Snyder (Eds.), *Biochemistry of Biogenic Amines* 381-442 (1975).
- [R33] Sijben, H. J., van Oostveen, W. M., Hartog, P. B. R., et al. Label-free high-throughput screening assay for the identification of norepinephrine transporter (NET/SLC6A2) inhibitors. *Sci. Rep.* **11**, 12290 (2021).
- [R34] Kelly, C. P., Cramer, C. J., & Truhlar, D. G. Adding explicit solvent molecules to continuum solvent calculations for the calculation of aqueous acid dissociation constants. *J. Phys. Chem. A* **110**, 2493-2499 (2006).
- [R35] Bergquist, J., Sciubisz, A., Kaczor, A., & Silberring, J. Catecholamines and methods for their identification and quantitation in biological tissues and fluids. *J. Neurosci. Meth.* **113**, 1-13 (2002).
- [R36] Bryantsev, V. S., Diallo, M. S., & Goddard, W. A., III. Calculation of solvation free energies of charged solutes using mixed cluster/continuum models. *J. Phys. Chem. B* **112**, 9709-9719 (2008).
- [R37] Lavanya, N., Fazio, E., Neri, F., Bonavita, A., Leonardi, S. G., Neri, G., & Sekar, C. Simultaneous electrochemical determination of epinephrine and uric acid in the presence of ascorbic acid using SnO₂ graphene nanocomposite modified glassy carbon electrode. *Sensor. Actuat. B-Chem.* **221**, 1412-1422 (2015).
- [R38] Sainz, R., del Pozo, M., Vilas-Varela, M., Castro-Esteban, J., Perez Corral, M., Vazquez, L., Blanco, E., Peña, D., Martín-Gago, J., Ellis, G., Petit-Domínguez, M., Quintana, C., Casero, E. Chemically synthesized chevron-like graphene nanoribbons for electrochemical sensors development: determination of epinephrine. *Sci. Rep.* **10**, 14614

(2020).

[R39] Yang, X., Zhao, P., Xie, Z., Ni, M., Wang, C., Yang, P., Xie, Y., Fei, J. Selective determination of epinephrine using electrochemical sensor based on ordered mesoporous carbon / nickel oxide nanocomposite. *Talanta* **233**, 122545 (2021).

[R40] Helmschrodt, C., Becker, S., Perl, S., Schulz, A., & Richter, A. Development of a fast liquid chromatography-tandem mass spectrometry method for simultaneous quantification of neurotransmitters in murine microdialysate. *Anal. Bioanal. Chem.* **412**, 7777-7787 (2020).

[R41] Becker, S., Schulz, A., Kreyer, S., Dreßler, J., Richter, A., & Helmschrodt, C. Sensitive and simultaneous quantification of 16 neurotransmitters and metabolites in murine microdialysate by fast liquid chromatography-tandem mass spectrometry. *Talanta* **253**, 123965-123965 (2023).

[R42] An, J., Shi, Y., Fang, J., Hu, Y., & Liu, Y. Multichannel ratiometric fluorescence sensor arrays for rapid visual monitoring of epinephrine, norepinephrine, and levodopa. *Chem. Eng. J.* **425**, 130595 (2021).

[R43] Azargoshasb, T., Parvizi, R., Navid, H. A., Parsanasab, G.-M., & Heidari, H. Versatile selective absorption-based optical fiber toward epinephrine detection. *Sensor. Actuat. B-Chem.* **372**, 132551 (2022).

[R44] Luo, C., Chen, Y., Gu, J., Cai, H., Lin, H., Jin, Z., & Huang, C. Activatable NIR Fluorescence Probe for Epinephrine Detection and Bioimaging Based on Anionic Heptamethine Cyanine. *Anal. Chem.* **96**, 9969-9974 (2024).

[R45] Hyman, A. L., Dempsey, C. W., Fontana, C., Richardson, D. E., Rieck, R. W., & Kadowitz, P. J. Pulmonary vascular responses to forebrain stimulation in the cat. *Circ.Res.* **63**, 493-501 (1988).

[R46] Murray, K., Cremin, M., Schreiber, S., Baumgarth, N., & Reardon, C. Afferent selective vagus nerve stimulation reduces TLR7 induced acute lung inflammation. *Physiology* **38**, S1 (2023).

[R47] Noble, L. J., Souza, R. R., & McIntyre, C. K. Vagus nerve stimulation as a tool for enhancing extinction in exposure-based therapies. *Psychopharmacology* **236**, 355-367 (2019).

[R48] Norcliffe-Kaufmann, L., Palma, J.-A., Martinez, J., Camargo, C., & Kaufmann, H. Fear conditioning as a pathogenic mechanism in the postural tachycardia syndrome. *Brain* **145**, 3763-3769 (2022).

[R49] Sousa, V. C., Mantas, I., Stroth, N., Hager, T., Pereira, M., Jiang, H., Jabre, S., Paslawski, W., Stiedl, O., Svenningsson, P. P11 deficiency increases stress reactivity along with HPA axis and autonomic hyperresponsiveness. *Mol. Psychiatr.* **26**, 3253-3265. (2021).

[R50] Azevedo, M., Martinho, R., Oliveira, A., Correia-de-Sa, P., & Moreira-Rodrigues, M. Molecular pathways underlying sympathetic autonomic overshooting leading to fear and traumatic memories: looking for alternative therapeutic options for post-traumatic stress disorder. *Front. Mol. Neurosci.* **16**, 1332348 (2024).

[R51] Killcross, S., Robbins, T. W., & Everitt, B. J. Different types of fear-conditioned behaviour mediated by separate nuclei within amygdala. *Nature* **388**, 377-380 (1997).

[R52] Penzo, M. A., Robert, V., Tucciarone, J., De Bundel, D., Wang, M., Van Aelst, L., Darvas, M., Parada, L. F., Palmiter, R. D., He, M., Huang, Z. J., Li, B. The paraventricular thalamus controls a central amygdala fear circuit. *Nature* **519**, 455-459 (2015).

[R53] Alves, E., Lukoyanov, N., Serrao, P., Moura, D., & Moreira-Rodrigues, M. Epinephrine increases contextual learning through activation of peripheral β_2 -adrenoceptors. *Psychopharmacology* **233**, 2099-2108 (2016).

[R54] Rudy, J. W., Huff, N. C., & Matus-Amat, P. Understanding contextual fear conditioning: insights from a two-process model. *Neurosci. Biobehav. Rev.* **28**, 675-685 (2004).

[R55] Toth, M., Ziegler, M., Sun, P., Gresack, J., & Risbrough, V. Impaired conditioned fear response and startle reactivity in epinephrine-deficient mice. *Behav. Pharmacol.* **24**, 1-9 (2013).

[R56] Do-Monte, F. H., Quinones-Laracuente, K., & Quirk, G. J. A temporal shift in the circuits mediating retrieval of fear memory. *Nature* **519**, 460-463 (2015).

[R57] Pessoa, L. A Network Model of the Emotional Brain. *Trends Cogn. Sci.* **21**, 357-371 (2017).

- [R58] Spencer, S. J., Fox, J. C., & Day, T. A. Thalamic paraventricular nucleus lesions facilitate central amygdala neuronal responses to acute psychological stress. *Brain Res.* **997**, 234-237 (2004).
- [R59] Xing, B., Li, Y. C., & Gao, W. J. Norepinephrine versus dopamine and their interaction in modulating synaptic function in the prefrontal cortex. *Brain Res.* **1641**, 217-233. (2016).
- [R60] Liu, C., & Kaeser, P. S. Mechanisms and regulation of dopamine release. *Curr. Opin. Neurobiol.* **57**, 46-53. (2019).
- [R61] Jamal, T., Yan, X., Lantyer, A. D. S., Ter Horst, J. G., & Celikel, T. Experience-dependent regulation of dopaminergic signaling in the somatosensory cortex. *Prog. Neurobiol.* **239**, 102630. (2024).